# Artificial photosynthetic cell producing energy for protein synthesis

Samuel Berhanu[1], Takuya Ueda[1] & Yutetsu Kuruma [2,3]

Attempts to construct an artificial cell have widened our understanding of living organisms. Many intracellular systems have been reconstructed by assembling molecules, however the mechanism to synthesize its own constituents by self-sufficient energy has to the best of our knowledge not been developed. Here, we combine a cell-free protein synthesis system and small proteoliposomes, which consist of purified ATP synthase and bacteriorhodopsin, inside a giant unilamellar vesicle to synthesize protein by the production of ATP by light. The photo-synthesized ATP is consumed as a substrate for transcription and as an energy for translation, eventually driving the synthesis of bacteriorhodopsin or constituent proteins of ATP synthase, the original essential components of the proteoliposome. The de novo photosynthesized bacteriorhodopsin and the parts of ATP synthase integrate into the artificial photosynthetic organelle and enhance its ATP photosynthetic activity through the positive feedback of the products. Our artificial photosynthetic cell system paves the way to construct an energetically independent artificial cell.

[1] Department of Computational Biology and Medical Sciences, Graduate School of Frontier Sciences, The University of Tokyo, Bldg. FSB-401, 5-1-5 Kashiwanoha, Kashiwa, Chiba 277-8562, Japan. [2] Earth-Life Science Institute, Tokyo Institute of Technology, 2-12-1-IE-1, Ookayama, Meguro-ku, Tokyo 152-8550, Japan. [3] JST, PRESTO, Saitama 332-0012, Japan. Correspondence and requests for materials should be addressed to T.U. (email: ueda@edu.k.u-tokyo.ac.jp) or to Y.K. (email: kuruma@elsi.jp)

Recent advances in synthetic biology allow us to challenge whole reconstruction of cell from simple non-living molecules and redesigned minimal genome[1–4]. Such attempts for the construction of artificial cell would lead not only to determining the necessary requirements for life phenomena but also to developing as a biodevice toward industrial application[5]. A cell-mimicking artificial cell is constructed by encapsulating a cell-free protein synthesis system inside giant vesicle. Cell-free system has been widely applied to researches in the field of synthetic biology, and especially a reconstructed cell-free system (PURE system)[6] has been used as a basic technology for the artificial cell construction because all constituent enzymes are known. This would be rather important when we try to reconstruct self-reproducing artificial cells that have to synthesize all their own components. Although several cellular functions or phenomenon have been reconstructed so far in the artificial cell system[7–12], an energy self-supplying system for the internal protein synthesis has not been achieved. To develop the artificial cell into the energetically independent system, it is necessary to set up a circulating energy-consumption and production system driven by an unlimited external physical or chemical energy source. For this purpose, a biomimetic artificial organelle producing adenosine triphosphate (ATP) by collaborating ATP synthase and bacteriorhodopsin is applicable as a rational energy generating system for artificial cells[13–18]. Recently, Lee et al.[18] performed ATP synthesis using similar photosynthetic artificial organelle, where they demonstrated carbon fixation (in vitro) and actin polymerization within giant unilamellar vesicle (GUV). This result evokes us to apply the artificial organelle into the artificial cell system, i.e., protein synthesis based on the photosynthesized ATP inside GUV. In this study, we performed ATP synthesis by light-driven artificial organelle inside GUV. Through optimization for the preparation method of proteoliposomes containing bacteriorhodopsin and ATP synthase, we succeeded to produce millimolar level ATP inside GUVs, wherein 4.6 µmol ATP per mg ATP synthase was produced after 6 h of illumination. By combining the artificial organelle and PURE system, we design and construct an artificial photosynthetic cell that produces ATP for the internal protein synthesis. The produced ATP was consumed as a substrate of messenger RNA (mRNA), or as an energy for aminoacylation of transfer RNA (tRNA) and for phosphorylation of guanosine diphosphate (GDP) (Fig. 1a and Supplementary Fig. 1). Additionally, we also demonstrated photosynthesis of bacteriorhodopsin or a membrane portion of ATP synthase, which is the original component of the artificial organelle. Our artificial cell system enables the self-constitution of its own parts within a structure of positive feedback loop.

## Results

**Construction of light-driven artificial organelle**. Light-driven artificial organelle was composed of two kinds of membrane proteins, bacteriorhodopsin (bR) and F-type ATP synthase ($F_oF_1$). bR was isolated from a purple membrane of *Halobacterium salinarum* by ultra-centrifugation with sucrose density gradient (Fig. 1b and Supplementary Fig. 2). $F_oF_1$ of *Bacillus* PS3 was purified as recombinant protein from *Escherichia coli* cells (Fig. 1b). The isolated bR were reconstructed as bR-embedding proteoliposomes (bR-PLs) for the measurement of light-dependent proton-pump activity. The size of bR-PLs were mostly 100–200 nm as diameter. We used phosphatidylcholine extract from soybean to form PLs which are stable in the reaction mixture of PURE system and also maintain the $F_oF_1$ activity[10]. The formation of bR-PLs was carried out by reducing the detergent concentration in the mixture of lipids and purified protein according to the previous report[19]; however, we have

found that only 25% bR were maintaining the proper membrane orientation (Supplementary Fig. 3C). To improve this ratio, we did some modifications in the preparation method by changing the timing of bR addition (Supplementary Fig. 3A), i.e., empty liposomes were first roughly preformed and, then, the purified bR was combined before completely removing the detergent. By this method, 70% bR was properly reconstructed in the PLs (Supplementary Fig. 3C). The improvement of the membrane orientation faithfully reflected into the proton-pump activity (Supplementary Fig. 3D). Since the efficiency of proton gradient generation directly affects the $F_oF_1$ activity, we employed this optimized method for all of the following experiments.

During the light illumination, we observed a decrease of proton concentration at the outside of bR-PLs in proportion to bR concentration (Fig. 1c), suggesting that the protons were transported from the outside to inside of the bR-PL lumen (Supplementary Fig. 1A). In addition to the proton-pump activity, we also observed a rapid return of the proton concentration when the illumination ceased. This indicates proton leakage from the inside to outside of the bR-PL lumen. The proton leak was accelerated when the lateral fluidity of the bR-PL membranes was increased by temperature rise (Supplementary Fig. 4). For the sake of inhibiting the leak through the membrane, we added 30% cholesterol into the lipid composition of bR-PLs[20], which resulted in 30% reduction of the proton leak (Supplementary Fig. 5). Thus, we kept this condition throughout the study.

Next, we estimated the membrane orientation of the reconstituted bR by evaluating the binding sensitivity of a histidine-tag, which elongated at the C-terminus of recombinant bR, to the Ni-NTA-conjugated magnet beads (Supplementary Fig. 6). If the reconstructed bR was keeping the working orientation, the C-terminus histidine-tag can bind to the magnet beads and be eluted in the elution fraction. The ratio of bR obtained in the elution fraction was normalized with the ratio of control experiment in which bR was monodispersed by dissolving the PLs with detergent (Triton). In the control experiment, 91% bR was collected in the elution fraction, although that should be 100% theoretically (Supplementary Fig. 6). Considering this result, we calculated that 86% bR was reconstructed in the working (outward C-terminus) orientation within the PL membrane; i.e., $Elu._{-Triton} Elu._{+Triton}{}^{-1}$ 100%. It should be noted that the opposite orienting bRs (inward C-terminus) pump protons from the inside to outside of the PLs. Thus, the net-working ratio of the reconstituted bR is calculated as 72% (Supplementary Table 1). Taking account of the bR membrane orientation, the initial reaction rate of bR was calculated as $-2.87 \pm 0.53$ $\Delta$pH min$^{-1}$ nmol$^{-1}$ or $-0.11 \pm 0.02$ $\Delta$pH min$^{-1}$ mg$^{-1}$, mean $\pm$ S.D. (Fig. 1c and Supplementary Table 1). On the other hand, the net-working ratio of the reconstituted $F_oF_1$ was 65.1% after the normalization as with bR (Supplementary Fig. 7 and Supplementary Table 1), and the initial reaction rate was $128 \pm 3.2$ ATP nmol min$^{-1}$ nmol$^{-1}$ or $223 \pm 6.1$ ATP nmol min$^{-1}$ mg$^{-1}$ (Fig. 1d and Supplementary Table 1). The reverse function of $F_oF_1$, ATP-dependent proton-pump activity, was also detected (Supplementary Fig. 8), suggesting the full functionality of the reconstituted $F_oF_1$-PLs.

To construct artificial organelle, we assembled purified bR and $F_oF_1$ to form $bRF_oF_1$-PLs. We prepared PLs in various proportion of bR against $F_oF_1$ and illuminated with visible light passing a 500 nm long-pass filter. The amount of produced ATP was measured by means of luciferin and luciferase. The highest ATP photosynthesis was obtained in the case of 176 µM bR and 1 µM $F_oF_1$. This means that approximately $0.6 \times 10^6$ ATP was produced by a single $bRF_oF_1$-PL within 4 h of illumination (Fig. 1e). The maximum turnover number for ATP synthesis in the initial 5 min

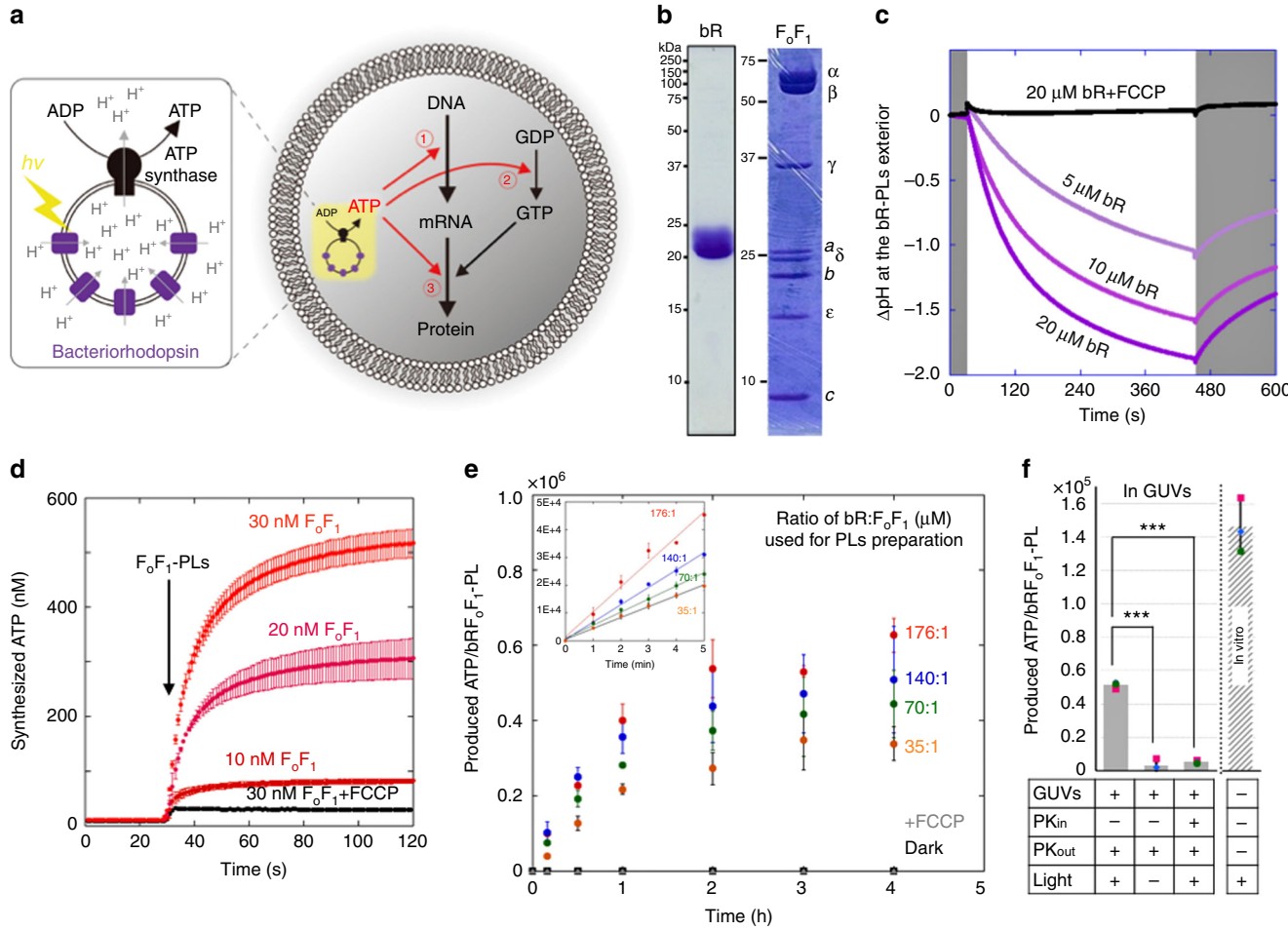

**Fig. 1** Light-driven adenosine triphosphate (ATP) synthesis by artificial organelle. **a** Schematics of the artificial photosynthetic cell encapsulating artificial organelle, which consists of bacteriorhodopsin (bR) and $F_oF_1$-ATP synthase ($F_oF_1$). Synthesized ATP are consumed as substrates for messenger RNA (mRNA) (①), as energy for phosphorylation of guanosine diphosphate (GDP) (②) or as energy for aminoacylation of transfer RNA (tRNA) (③). **b** Sodium dodecyl sulfate-polyacrylamide gel electrophoresis (SDS-PAGE) analysis of purified bR and $F_oF_1$. The positions of molecular markers and $F_oF_1$ component proteins are indicated beside the gels. **c** Light-driven proton-pump activity of bR reconstituted in a proteoliposome (PL). Proton-pump activity of bR was measured by monitoring the proton concentration at the outside of bR-PLs where fluorescent proton-sensor ACMA (9-amino-6-chloro-2-methoxy acridine) was added. We defined as $\Delta$pH = pH (original, outside) − pH (after illumination, outside). The $\Delta$pH caused by bR activity was measured with the various bR concentrations as indicated. White and gray areas indicate light ON and OFF, respectively. An uncoupler, FCCP (carbonyl cyanide 4-(trifluoromethoxy) phenylhydrazone), was used as a control experiment. **d** ATP synthesis activity of $F_oF_1$ reconstituted as $F_oF_1$-PLs. ATP synthesis reactions were initiated by adding $F_oF_1$-PLs at 30 s with various $F_oF_1$ concentrations, as indicated. The synthesized ATP was measured by means of luciferin and luciferase (see Methods section for the experiment details). FCCP was used for control. **e** Light-driven ATP synthesis. The amount of the photosynthesized ATP by bR$F_oF_1$-PLs, which was constituted in various proportions of bR against $F_oF_1$, were measured by luciferin and luciferase. FCCP and dark conditions were also performed as controls. The inset indicates initial rate of the each PL. **f** Light-driven ATP synthesis inside giant unilamellar vesicle (GUV). bR$F_oF_1$-PLs were illuminated inside GUVs in the presence or absence of proteinase K (PK) that degrades the $F_oF_1$. The in vitro experiment was also performed for comparison. ***$p < 0.001$. P values were from two-sided $t$-test. All experiments were repeated at least three times, and their mean values and standard deviations (S.D.) are shown. Source data are provided as a Source Data file

was $8.3 \pm 0.3\ s^{-1}$ in the case of 176 μM bR and 1 μM $F_oF_1$. This was almost double compared to the previous report[18]. Here, in a single PL, 3560 of the working bRs drive 18 $F_oF_1$ (Supplementary Table 1). In all cases, we used 10 mM $NaN_3$ to inhibit the reverse (ATPase) activity of $F_oF_1$[21]. We found that the ATP production plateaued when the illumination was higher than 10 mW per $cm^2$ (Supplementary Fig. 9).

The same reaction was also performed inside GUVs in which about $1.1 \times 10^4$ bR$F_oF_1$-PLs are contained in a 10 μm diameter GUV. After 6 h of illumination, we observed photosynthesized ATP from the inside of the GUVs (Fig. 1f), where 1.8 mM ATP was produced in a single GUV (Supplementary Table 1). This represents that 4.6 μmol ATP was produced per mg ATP

synthase. The efficiency of ATP production in GUVs was roughly one-third that of the in vitro system, perhaps caused by lower light intensity inside a GUV. Since our artificial organelle can produce ATP inside GUV at the comparable concentration as a real living cell, we proceeded to design and construct the photosynthetic artificial cell system that synthesize protein by light.

**Light-driven protein synthesis inside the artificial cell.** We performed green fluorescent protein (GFP) synthesis inside GUVs by means of the photosynthesized ATP to demonstrate that the constructed artificial organelle works in an artificial cell

system. For this purpose, we combined $bRF_oF_1$-PLs with the PURE system which is a cell-free protein synthesis system. The PURE system is reconstituted from purified translation factors[6], and therefore we can customize the component factors suited for the designed artificial system. The PURE system was modified as shown in Supplementary Table 2 to allow the photosynthesized ATP be specifically used for the aminoacylation of tRNA (Supplementary Fig. 10), and supplied with a mRNA encoding GFP together with $bRF_oF_1$-PLs and $NaN_3$. $NaN_3$ did not inhibit protein synthesis at concentrations below 50 mM (Supplementary Fig. 11). The prepared reaction mixture was encapsulated inside GUVs, and illuminated to induce protein synthesis. A large majority of the GUV population appeared in a range of 10–20 μm as diameter ($n = 200$) (Supplementary Fig. 12). After 6 h, we observed the fluorescence of internally synthesized GFP by confocal microscopy (Fig. 2a). This GFP synthesis was also

confirmed in vitro (without GUVs) (Fig. 2b and Supplementary Fig. 13) and synchronized with timing of the ATP photosynthesis (Supplementary Fig. 14). These results indicate that GFP synthesis inside the GUVs was driven by the photosynthesized ATP (Supplementary Fig. 1B). We found that 50–60% of the GUVs emitted more fluorescence than the nonilluminated control GUVs (Fig. 2c) by flow-cytometry analysis. We also found a certain percentage of GUVs were not showing fluorescent even when illuminated. Although the definitive cause is not unclear, it has been reported that the encapsulation efficiency of PLs is affected by the size of PLs; i.e., less than 35% GUVs can encapsulate the PLs when their size are over 200 nm (diameter)[18]. Additionally, we cannot deny the possibility that inactivity of the internal artificial organelle by the fusion of $bRF_oF_1$-PLs and GUV membranes is limiting the successful artificial cell formation.

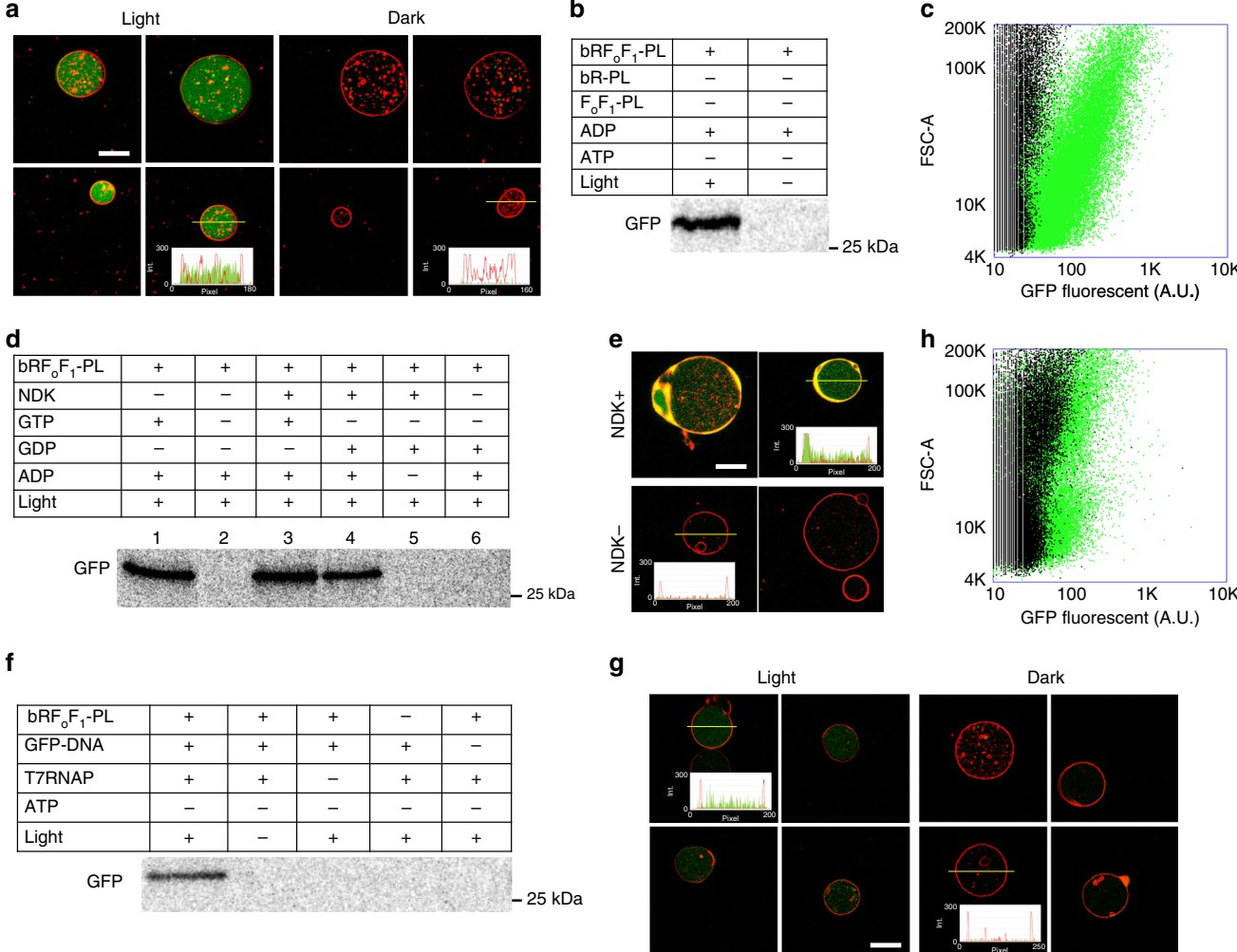

**Fig. 2** Protein synthesis inside giant unilamellar vesicle (GUV) driven by light. Green fluorescent protein (GFP) was synthesized from its messenger RNA (mRNA) (**a**–**e**) or DNA (**f**–**h**) inside light illuminated GUV (**a**, **c**, **e**–**h**) or in vitro (**b**, **d**). GFP was synthesized inside GUV (**a**) or in vitro (the PURE system) (**b**) in which the photosynthesized adenosine triphosphate (ATP) was consumed for the aminoacylation of transfer RNA (tRNA). The insets in **a**, **e** and **g** indicate plot profile of green and red colors on the thin yellow line. **c** Flow-cytometric analysis of the GUVs of **a**. The illuminated GUVs are shown as green, whereas the GUVs incubated in the dark are shown as black. The *X*- and *Y*-axes represent the fluorescent intensity and the area of forward scattering, respectively. **d** GFP synthesis coupled with guanosine 5′-triphosphate (GTP) generation. GFP was synthesized in the PURE system with or without nucleoside-diphosphate kinase (NDK), GTP, guanosine diphosphate (GDP) and adenosine 5′-diphosphate (ADP). **e** The same reactions as in lanes 4 and 6 of **d** were performed inside GUVs as indicated as NDK+ and NDK−, respectively. **f** GFP synthesis from its DNA. A gene of whole GFP was introduced in the PURE system with or without $bRF_oF_1$-PLs, T7 RNA polymerase (T7RNAP), ATP and light. **g** A small part of GFP (GFP11: 15 amino acids) was synthesized from its encoding DNA inside GUVs containing T7RNAP, another large part of GFP (GFP1-10) purified form *E. coli* cells, and the PURE system lacking NDK. **h** The same GUVs of **g** were analyzed by flow-cytometer as in **d**. The synthesized GFP in **b**, **d**, and **f** were labeled with [³⁵S] methionine. Scale bar: 10 μm. Source data are provided as a Source Data file

Next, we omitted GTP from the reaction mixture but introduced GDP and nucleoside-diphosphate kinases (NDKs), which allows the photosynthesized ATP to be consumed for synthesis of GTP that is a direct energy source of translation (Supplementary Fig. 10). The results showed that synthesized GFP was clearly detected by the sodium dodecyl sulfate–polyacrylamide gel electrophoresis (SDS-PAGE) analysis when adenosine 5'-diphosphate (ADP), GDP and NDK were added (Fig. 2d, lane 4). We performed the same reaction inside GUVs and observed the fluorescence emission from the GUV lumen (Fig. 2e), suggesting that the photosynthesized energy was consumed not only for aminoacylation of tRNAs but also directly for translation inside GUVs.

In real cells, ATP is consumed not only as energy but also as a substrate of transcription. To build up this, we performed a transcription-and-translation coupled reaction in the artificial photosynthetic cell system. When T7 RNA polymerase and template DNA coding-GFP were introduced into the PURE system, photosynthesis of GFP was clearly detected by SDS-PAGE analysis (Fig. 2f). However, we could not detect significant fluorescence by microscopy observation and flow-cytometer analysis when we performed inside GUVs. This is because the synthesized GFP level was lower than the detection limit. To overcome this problem, we applied the split-GFP method developed by Cabantous et al.[22], i.e., GFP is split into two parts: a small peptide (GFP11) and another large partner protein (GFP1-10). The fluorescence of GFP1-10 was restored by incorporating GFP11 (Supplementary Fig. 15). Although the intensity was rather weak, we observed the emission of GFP fluorescence from the GUVs when GFP11 was photosynthesized from the template DNA (Fig. 2g). In this reaction, we encapsulated the PURE system modified for transcription-and-translation reaction (Supplementary Table 2), and the purified GFP1-10. The successful photosynthesis of GFP11 was also confirmed in an in vitro reaction (Supplementary Fig. 16). By flow-cytometry analysis, we found that about 15% of the total GUVs emitted significant fluorescence as a consequence of transcription and translation inside (Fig. 2h). These results show that the photosynthesized ATP was consumed both as the substrates for mRNA transcription and as the energy for protein translation, just as in real cells.

**Self-production of the artificial organelle components**. The two kinds of component proteins of the artificial organelle produced ATP, and the resulting ATP drove protein synthesis. To test whether our artificial photosynthetic cell system can synthesize the component proteins of its own artificial organelle, we tried to photosynthesize bR, as well as $F_oF_1$. In this reaction, we used the translation-only PURE system (see For mRNA start in Supplementary Table 2). We expected that the newly photosynthesized de novo bRs localize onto the $bRF_oF_1$-PL membrane and increase ATP photosynthesis activity of the artificial organelle as a consequence of activity enhancement in the proton gradient generation of the $bRF_oF_1$-PL (Fig. 3a and Supplementary Fig. 1E and F). The fluorescence of the synthesized bR, which fused with GFP (bR-GFP), was mostly homogeneously observed inside the GUV lumen but not on the GUV membrane (Fig. 3b), indicating that the synthesized bR-GFP localized onto the internal PL membrane. This directed membrane localization is controlled by means of cholesterol which inhibits spontaneous membrane integration of protein[23]. We added 40% (mol%) cholesterol in the lipid composition of GUV membrane but not in the internal PL membrane. When bR-GFP was synthesized inside GUVs containing liposomes, the same homogeneous fluorescent distribution was observed within the GUV lumen. In contrast, when

the GUVs were not encapsulating liposomes, many larger-size puncta appeared, suggesting aggregation of the synthesized bR-GFP (Supplementary Fig. 17). These results imply that the bR-GFP synthesized in the GUVs (Fig. 3b) localized onto the internal PL membrane avoiding protein aggregation. The membrane localization of bR was further confirmed in vitro by flotation assay. When bR was synthesized in a standard PURE system in the presence of liposomes, we found that the synthesized bR appeared in the liposome fractions (Fig. 3c) after ultra-centrifugation with a sucrose cushion, whereas almost all bR appeared in the pellet fraction when liposomes were omitted. This result directly shows the membrane localization of the de novo bR onto the PL membrane. Additionally, the membrane localized bR showed the proton-pump activity in response to the duration of protein synthesis reaction (Fig. 3d). Here, 11, 61, 124 or 233 bRs per one liposome were synthesized at the time of 10, 30, 60 or 180 min reaction, respectively (Supplementary Fig. 18). These results lead us to conclude that the de novo photosynthesized bRs spontaneously localized onto the internal PL membrane and may have increased the proton-pump activity there.

If the de novo photosynthesized bRs are functionable on the PL membrane, the ATP production rate of PL should be enhanced according to the increase of the number of bR per PL. To confirm this, we measured ATP concentration in the PURE system reaction mixture during the photosynthesis of de novo bR ($bR_{wt}$). In this experiment, we used the PLs consisting of a low concentration bR (i.e. 5 μM bR) to emphasize the effects of the de novo photosynthesized bR. The effect of the de novo photosynthesized bR was determined by comparing to the control experiment synthesizing a mutant bR ($bR_{mut}$) which does not have any proton-pomp activity (Supplementary Fig. 19), and therefore the ATP production rate of the PLs containing $bR_{mut}$ is constant throughout the bR photosynthesis. In the case of $bR_{wt}$ photosynthesis, the ATP concentration was higher than that in the case of $bR_{mut}$ photosynthesizing in all three independent measurements (Supplementary Fig. 20), especially after 10 min reaction. This is consistent with the result of proton-pump activity of the bR synthesized in PURE system (Fig. 3d). Here, the difference in the ATP concentration between $bR_{wt}$ and $bR_{mut}$ photosynthesizing reactions represents the effect of de novo photosynthesized $bR_{wt}$. The ATP concentration in the $bR_{wt}$-photosynthesizing reaction was approximately 1.5-fold higher than that of $bR_{mut}$ (Supplementary Fig. 20). It should be noted that the obtained ATP concentration indicates the net of the photosynthesized ATP minus the consumed ATP for the protein synthesis. The synthesis rate of the $bR_{wt}$ and $bR_{mut}$ was adjusted to be the same by regulating the amount of template mRNA (Supplementary Fig. 21), and thus the ATP consumption rates were equal in both cases. To directly compare all three measurements, we normalized each obtained result with the ATP per PL value at the endpoint time (120 min) of the de novo $bR_{mut}$-photosynthesizing reaction. The ratio of the increased artificial organelle activity is shown in Fig. 3e. We also confirmed the photosynthesized bR by SDS-PAGE analysis (Fig. 3f) in which the photosynthesis rate was 2.5 nmol ml$^{-1}$ min$^{-1}$. After 2 h of reaction, the number of working bRs per PL increased from 100 working bRs per PL (original) to 110 bRs per PL (after photosynthesis). These series of results indicate that the ATP production rate was enhanced during the photosynthesis of de novo $bR_{wt}$ because the ability of proton gradient generation was improved by increasing the number of functional bRs on a PL.

Finally, we challenged to photosynthesize de novo $F_oF_1$ in vitro and to observe the enhancement of ATP production activity of the resulting PLs. Unlike bR, $F_oF_1$ consists of eight kinds of subunit proteins. Thus, we first try to synthesize these eight kinds

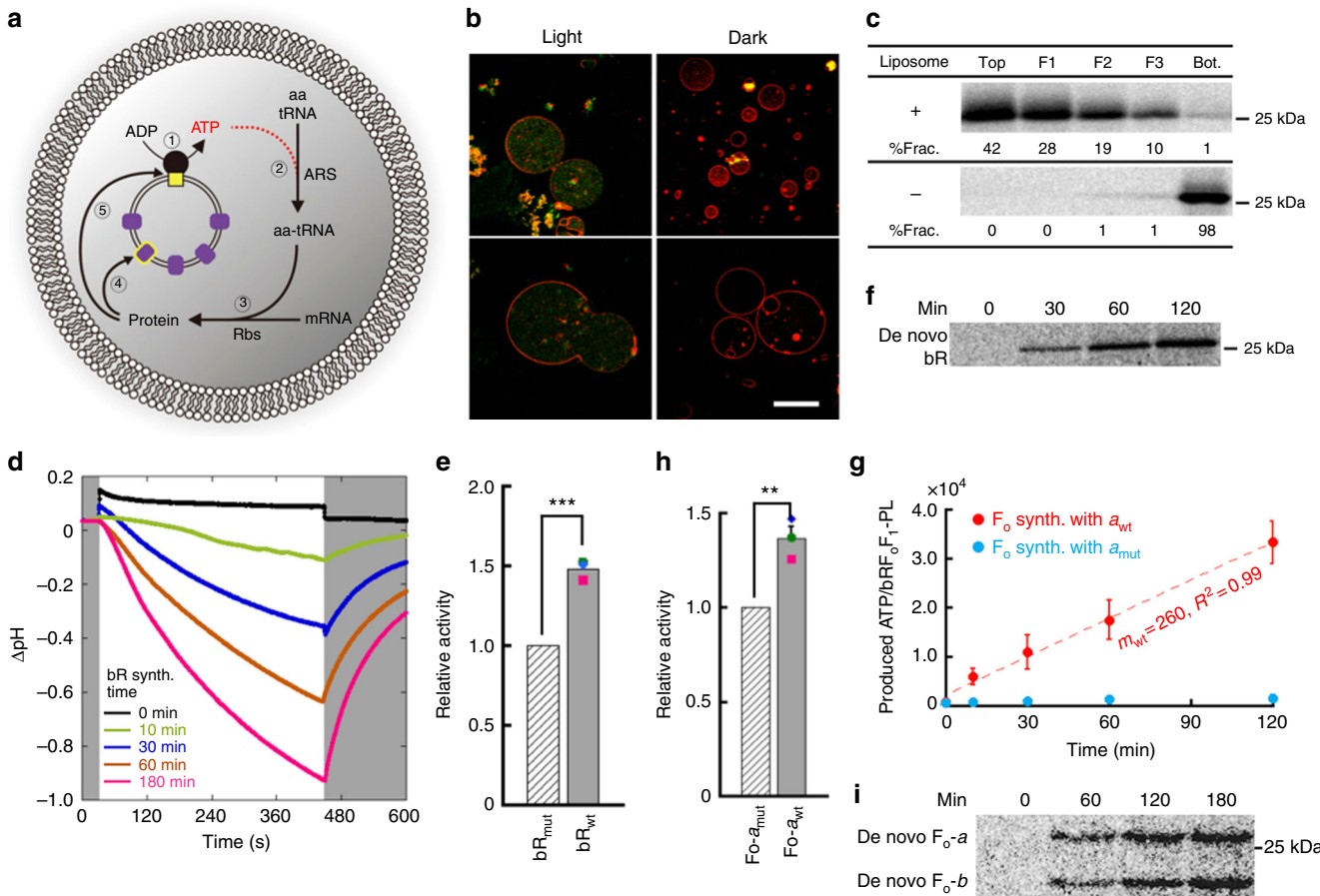

**Fig. 3** Self-constituting protein synthesis in artificial photosynthetic cells. **a** Schematics of self-constituting protein synthesis. The numbers indicate the order of reactions; ①: adenosine triphosphate (ATP) synthesis, ②: aminoacylation of transfer RNA (tRNA) by aminoacyl-tRNA synthetase (ARS), ③: translation by ribosomes (Rbs), ④: de novo bacteriorhodopsin (bR) synthesis, and ⑤: de novo $F_o$ synthesis. **b** Light-induced bR-GFP synthesis in giant unilamellar vesicles (GUVs). Bar: 10 μm. **c** Membrane localization of bR. The bRs synthesized in the PURE system with or without liposomes were fractionated by ultra-centrifugation with sucrose cushion. The percentages of bR in each fraction (%Frac.) are indicated at the bottom of the gels. **d** Proton-pump activity of bR synthesized in the PURE system. The measurement was performed as in Fig. 1c. The reaction times of protein synthesis are indicated by different colors. The white and gray areas represent light ON and OFF, respectively. **e** Enhanced artificial organelle by de novo bR. Wild-type ($bR_{wt}$) or mutant ($bR_{mut}$) bRs were photosynthesized in the PURE system containing $bRF_oF_1$-PLs. The ATP concentrations at the time 2 h was measured and converted into ATP per proteoliposome (PL). The value of the de novo $bR_{wt}$-containing PL was normalized to that of the de novo $bR_{mut}$-containing PL. \*\*\*$P < 0.001$. **f** Sodium dodecyl sulfate–polyacrylamide gel electrophoresis (SDS-PAGE) analysis of the de novo photosynthesized bR. **g** Light-driven ATP synthesis by PLs consist of cell-free synthesized $F_o$. Wild-type ($a_{wt}$) or mutant ($a_{mut}$) a-subunit protein was synthesized together with b- and c-subunits in the PURE system containing purified $F_1$ and bR-PLs. The measured ATP concentrations were converted into the produced ATP per PL. **h** Enhanced artificial organelle by de novo $F_o$. $a_{wt}$ or $a_{mut}$ was photosynthesized together with b- and c-subunits in the presence of purified $F_1$ and $bRF_oF_1$-PLs. The ATP concentrations at the time 3 h was measured and converted into ATP per PL. The value of the de novo $a_{wt}$-containing PL was normalized by that of the de novo $a_{mut}$-containing PL. \*\*$P < 0.01$. **i** SDS-PAGE analysis of the de novo photosynthesized $F_o$. $P$ values were from two-side $t$-test. All experiments were performed at least three times and their means and S.D. are shown. Source data are provided as a Source Data file

of proteins by adding their corresponding template DNAs into a standard PURE system supplemented with liposomes. However, unfortunately, we could not detect a significant activity of the $F_oF_1$ due to low yields. We next synthesized only three component proteins of $F_o$, a-, b- and c-subunits, in the presence of purified $F_1$ and bR-PLs. After the reaction, the resulting $bRF_oF_1$-PLs were isolated from the reaction mixture and illuminated with supplying ADP. The result shows that ATP photosynthesis of the PLs was detected in proportion to illumination time, when wild-type a-subunit ($a_{wt}$) protein was synthesized (Fig. 3g) with other b- and c-subunit proteins. This indicates the cell-free synthesized $F_o$ localized onto the bR-PL membrane and photosynthesized ATP by co-working with bR. Contrary, we could not detect any activity when a mutant a-subunit ($a_{mut}$), which has an amino acid substitution at R169 to

alanine[24], was synthesized instead of the wild-type a. This further supports that the cell-free synthesized $F_o$ formed functional $F_oF_1$ onto the PL membrane and synthesized ATP, and thus we next tried to photosynthesize $F_o$ and observed the enhancement of ATP photosynthesis activity in the resulting $bRF_oF_1$-PLs. The photosynthesis reaction of $F_o$ was performed in the translation only system. The a-, b- and c-subunit proteins form the complex structure of $F_o$ in the stoichiometry of 1, 2 and 10, respectively. In order to find the best proportion of these three templates for obtaining the highest $F_oF_1$ activity, we tested various proportions of the template DNA mix, at first. The multi-protein synthesis for $F_o$ was performed in the presence of liposomes and purified $F_1$. We detected the highest $F_oF_1$ activity when 4, 2 and 10 nM template DNA of a-, b- and c-subunit, respectively, were added (Supplementary Fig. 22). Following this, we photosynthesized de

novo $F_o$ component proteins in the presence of $bRF_oF_1$-PLs (0.3 μM $F_oF_1$ and 140 μM bR) and purified $F_1$ in the PURE system suited for mRNA start (Supplementary Table 2 and Supplementary Fig. 1F). The $F_o$ photosynthesis rate was 50 fmol $ml^{-1}$ $min^{-1}$ and reached to the plateau level at 7 h (Supplementary Fig. 23). After the $F_o$ photosynthesis reaction, PLs were isolated from the reaction mixture and illuminated in the presence of ADP. In order to distinguish the effect of de novo photosynthesized $F_o$, we also synthesized $a_{mut}$ instead of $a_{wt}$ and compared them, same as in the case of de novo bR photosynthesis. The PL-containing de novo $F_o$-$a_{wt}$ (PLs-$a_{wt}$) showed higher ATP photosynthetic activity than the PL-containing de novo $F_o$-$a_{mut}$ (PLs-$a_{mut}$) in all three independent measurements (Supplementary Fig. 24A). The enhancement of ATP photosynthesis rate per PL was 1.38-fold. Since we recovered the PLs-$a_{wt}$ and PLs-$a_{mut}$ from the reaction mixtures, the amount of PLs analyzed was same in both samples (Supplementary Fig. 24B), and therefore the difference in activities of PLs-$a_{wt}$ and PLs-$a_{mut}$ is thought to be reflecting the enhanced activity by the de novo photosynthesized $F_o$. We also confirmed the same amount of $F_o$ component proteins were photosynthesized in both samples (Supplementary Fig. 24C). Based on these results, we analyzed the enhanced ATP photosynthesis rate in the resulting PLs. After 7 h of photosynthesis, the net concentration of de novo photosynthesized $F_o$ within the reaction mixture was 20 nM (Supplementary Fig. 23). Since 18 nM PLs were contained in the reaction mixture, statistically one de novo $F_o$ was assigned to one PL. This reflected into the enhanced ATP photosynthesis rate of the PL as 101.4 ATP $PL^{-1}$ $min^{-1}$ (Supplementary Fig. 24A), which represents 101.4 ATP $F_oF_1^{-1}$ $min^{-1}$ (turnover number: 1.7 $s^{-1}$). The calculated result consistent to the specific activity of $F_oF_1$ reconstructed into $F_oF_1$-PLs, 118±3.2 nmol ATP $min^{-1}$ $nmol^{-1}$ (Supplementary Table 1). On the other hand, when PLs-$a_{mut}$ was photosynthesized, the ATP photosynthesis rate showed 268 ATP $PL^{-1}$ $min^{-1}$. Since five working-$F_oF_1$ were contained in one PL, it can be converted as about 50 ATP $F_oF_1^{-1}$ $min^{-1}$ that is lower than the de novo $F_o$ activity which we cannot explain well. Overall, the obtained result of the $F_o$ photosynthesis seems reasonable. As we showed above, recursive production of $F_o$ portion of $F_oF_1$ was definitely observed, though it did not enhance exponentially. Although the photosynthesis level is still low, we engineered a self-constituting protein synthesis positive feedback loop in the artificial photosynthetic cells.

## Discussion

We show that our artificial cell system containing the artificial organelle was able to first transduce light energy into an electrochemical potential, and then convert into the chemical energy of ATP inside GUV. The produced ATP was consumed for the reaction of aminoacylation of tRNA as well as for the generation of GTP which was eventually consumed for translation. We also showed that the produced ATP was converted into a mRNA that subsequently translated into a part of GFP. The biochemical reactions performed in our artificial cell system mimic that is occurring in real living cells. Finally, we performed the photosynthesis of bR and $F_o$. The photosynthesized de novo bR localized onto the membrane of internal artificial organelle and enhanced the activity of ATP production, indicating the functional engagement of protein synthesis and energy production reactions. Because bR is the original compound of the artificial organelle, we demonstrated that the artificial cell synthesized its own part in a positive feedback loop. Furthermore, another membrane-embedding component, $F_o$, was photosynthesized and its functional contribution in ATP photosynthesis was detected. It should be noted that all these reactions were reconstructed with a minimal number of enzymes and molecules since we used a reconstructed artificial organelle and cell-free protein synthesis system[6]. The functional significance in our artificial cell would accelerate the researches of artificial cell (or synthetic cell) in the field of synthetic biology, as well as the development of a biodevice sensing light and promoting protein and RNA synthesis. For example, our artificial cell technique would be applicable into the study of drug delivery that can control spatiotemporal production of aptamer or single chain Fv within a vesicle capsule. More promising application of the artificial organelle is to use as the phosphate recycling system in cell-free system. The current cell-free system is using creatine phosphate as a primary energy source; however, since this is unidirectional reaction, free phosphates accumulate in the system as the reaction goes on. Our artificial organelle can avoid this problem by recharge the free phosphate onto ADP after the ATP consumption.

Artificial cells have been employed as a model of protocell or primordial cell, which are thought to have existed before modern cells, in the study of origin of life[2,18,25–28]. Especially, how the primordial cell gained the ability to produce an energy to drive primitive metabolism is a big argument[29]. The genes of ATP synthase are highly conserved beyond the species and have been thought to exist from early stage of life[30]. However, what mechanism generated a proton gradient to drive ATP synthase before the completion of the complicated electron transfer system is still unknown. Our work demonstrated that a simple bio-system, which consists of two kinds of membrane proteins, is able to supply sufficient energy for operating gene expression inside a microcompartment. Thus, we think that primordial cells using sunlight as a primal energy source could have existed in the early stage of life before evolving into an autotrophic modern cell system. We believe the attempts to construct living artificial cell will reveal the boundary state of the transition from non-living to living matters that actually happened in the early Earth environment.

## Methods

**Materials.** All reagents utilized in experiments were of the highest purity and grade. These include POPC (1-palmitoyl-2-oleoyl-*sn*-glycero-3-phosphocholine), cholesterol, PEG2000PE (1,2-distearoyl-*sn*-glycero-3-phosphoethanolamine-*N*-[methoxy (polyethylene glycol)-2000] (ammonium salt)), soybean phosphatidylcholine (SoyPC) extract; liquid paraffin (Wako); pH gradient sensitive fluorophore ACMA (9-amino-6-chloro-2-methoxy acridine), protonophore FCCP (carbonyl cyanide 4-(trifluoromethoxy) phenylhydrazone), ADP (adenosine 5'-diphosphate monopotassium salt), $P^1,P^5$-di(adenosine-5') pentaphosphate pentasodium salt (AP5A), potassium-specific ionophore valinomycin, Triton X-100, all-*trans* retinal (ATR) (Sigma); ATP (adenosine 5'-triphosphate), GTP (guanosine 5'-triphosphate), CTP (cytidine 5'-triphosphate), UTP (uridine 5'-triphosphate) (geneACT inc); OG (octyl β-D-glucopyranoside); DDM (*n*-dodecyl-β-D-maltoside) (Dojindo); Tween-20 (Calbiochem); sodium cholate (Wako); and ATP bioluminescence assay kit CLS II (Roche).

**Isolation of purple membrane.** Purple membrane patches containing firmly packed two-dimensional crystals of bR were isolated from *Halobacterium salinarum* R1 following the previous protocol[31] with slight modification. In brief, the *H. salinarum* colonies were cultured in 6 L tryptone media containing 4 M NaCl under high oxygen tension and continuous illumination with 200 W LED lamp for almost 192 h at 39 °C. The cells were then collected and resuspended in basal salt in the presence of 17 μg $ml^{-1}$ of DNaseI (Sigma-Aldrich). This was followed by overnight dialysis against 100 mM NaCl at 4 °C. The translucent red lysate was centrifuged at 45,000 × *g* for 60 min, and the precipitate was washed 6 times with 100 mM NaCl solution and distilled water. Eventually, the purple membrane precipitate was overlaid on a 30–50% linear sucrose gradient and centrifuged at 100,000 × *g* for 17 h using SW28 rotor. At the end of the centrifugation, the purple band was collected. The sucrose solution was then removed by centrifuging the purple suspension at 45,000 × *g* for 1 h. The purple membrane sediment was resuspended and later stored in 50 mM Tris-HCl (pH 7.6), 150 mM NaCl and 10% glycerol. Stock concentration of the purified bR was 450 μM.

**Overexpression and purification of bR from *E. coli*.** C43(DE3) *E. coli* strain was transformed with pET21c-bR construct (see the DNA sequence in Supplementary

Table 5). The resulting recombinant bR was bearing hexa-his-tag at its C-terminus. The bacterial culture was set in 6 L 2× YT media at 37 °C with shaking. The incubation continued until OD 0.6–0.7 and the culture media were supplemented with 1 mM isopropyl β-D-1-thiogalactopyranoside (IPTG) and 10 μM ATR. The incubation was continued for 1 h at 30 °C and then for 3 h at 37 °C. The cells were collected and washed. The collected cells were disrupted by 5 passes of French press homogenizer at 550 bar in a lysis buffer (50 mM MES (pH 6.0), 1 mM EDTA, 300 mM NaCl, proteinase inhibitor cocktail). The membrane fractions were collected by centrifuging the pure lysate at 234,788 × g for 1 h. The collected membrane fraction was solubilized overnight at 4 °C in a buffer containing 50 mM MES (pH 6.0), 300 mM NaCl, 5 mM imidazole and 1.5% DDM. The solubilized bR was purified in Ni-affinity chromatography (pre-equilibrated in a buffer containing 50 mM MES (pH 6.0), 300 mM NaCl, 0.2% DDM and 40 mM imidazole). The protein was eluted in a linear imidazole gradient of 40–300 mM. Further purification was done using Mono-Q column in the presence of 0.2% DDM. Finally, bR was eluted with linear NaCl gradient of 10–300 mM. Stock concentration of the purified recombinant bR was 1 μM.

**Expression and purification of recombinant $F_oF_1$ and $F_1$.** Thermophilic *Bacillus* PS3 $F_oF_1$-ATP synthase ($F_oF_1$) was overexpressed in DK8 *E. coli* strain (*unc* minus) carrying pTR19ASDSεΔc construct[24]. The culture was made in 3 L of 2× YT media for 21 h at 37 °C. The purification of $F_oF_1$ was undertaken in accordance with previous work[32] with modification. The cells were disrupted by a tip-sonication in a buffer containing 10 mM HEPES-KOH (pH 7.5), 5 mM MgCl₂, 10% Glycerol and 28 mM β-mercaptoethanol. The cell debris was removed by centrifuging the lysate at 5500 × g for 30 min at 4 °C accompanied by collecting the membrane faction containing $F_oF_1$ complex by ultra-centrifugation at 225,000 × g for 1 h. The membrane fraction was homogenized in buffer I (pH 7.5) (10 mM HEPES-KOH, 5 mM MgCl₂, 2% Triton X-100, 0.5% cholate, 10% glycerol and protease inhibitor cocktail) and incubated at 30 °C for 30 min with mild shaking. The sample was then centrifuged at 311,000 × g for 20 min at 30 °C and supernatant was collected. To the supernatant, 70 ml of buffer II (pH 7.5) (20 mM potassium phosphate, 100 mM KCl, 24 mM imidazole and protease inhibitor cocktail) was added. This was later applied to Ni-NTA (Qiagen) pre-equilibrated with buffer A (pH 7.5) (20 mM potassium phosphate, 100 mM KCl, 20 mM imidazole and 0.15% DDM). Finally, the $F_oF_1$ was eluted with buffer A (pH 7.5) containing 200 mM imidazole. The homogeneity and purity of the protein was judged by 10–20% gradient SDS-PAGE. Finally, pure and homogenous fractions were collected and buffer was exchanged with a buffer containing 20 mM KPi (pH 7.6), 100 mM NaCl and 0.15% DDM. This was concentrated eventually by 50 kDa Amicon. Stock concentration of the purified $F_oF_1$ was 99 μM.

The $F_1$ complex was purified following the previous literature by Suzuki et al.[33]. Briefly, *E. coli* cells expressing $F_oF_1$ were disrupted and the cytosol fraction was obtained by centrifugation. The supernatant was incubated at 67 °C for 15 min, and the aggregated *E. coli* proteins were removed by centrifugation. The resulting yellow supernatant was subjected to a Ni-NTA column. The eluted $F_1$ was incubated for 60 min at 25 °C, then ammonium sulfate was added to the solution. The resulting solution was applied to a phenyl-Toyopearl column. After washing, the column was eluted with a linear reverse gradient of ammonium sulfate (1–0 M), then fractions containing $F_1$ were collected and precipitated with ammonium sulfate. The $F_1$ was further purified with a Superdex 200HR column. The purified protein was stored at −80 °C. The stock concentration of $F_1$ was 5.7 μM.

**Preparation of split-GFP.** The split-GFP was prepared as previously described[22]. The split-GFP was prepared firstly by introducing 7 point mutations into the gene of superfolder-GFP (sfGFP), then by dividing into two portions. We used P1–P14 primer sets (Supplementary Table 3) to introduce point mutations N39I, T105K, E111V, I128T, K166T, I167V and S205T, by quick change PCR. The DNA strand for a larger part (GFP1-10) of the split-GFP was amplified by PCR using P15 and P16 primes and inserted into pET29a vector between the restriction enzyme sites of NdeI and EcoRI using infusion cloning technique. Thereby, a hexa-histidine-tag was introduced at the N-terminus of the open reading frame.

The resulting construct, pET29aGFP1-10 (see the DNA sequence in Supplementary Table 5), was introduced into BL21(DE3) strain for overexpression and purification of GFP1-10.

The *E. coli* strain B21(DE3) harboring pET29aGFP1-10 was cultured in LB media containing 50 μg ml⁻¹ Kanamycin. The culture was incubated under shaking at 37 °C until OD 0.6. Then, 1 mM IPTG was added for induction and the incubation continued for additional 6 h at 30 °C. The cells were then collected and washed one time. The collected cells were sonicated being suspended in a buffer containing 20 mM Tris-HCl (pH 8.0), 200 mM NaCl, 0.005 % Triton X-100 and protease inhibitor cocktail. After removing debris by centrifugation, the lysate was injected to His-Trap Ni-column which was pre-equilibrated with buffer A (pH 8.0) (20 mM Tris-HCl, 20 mM NaCl and 10 mM imidazole). After the washing, the protein was eluted with linear gradient of 10–300 mM of imidazole. Further purification was carried out using anion exchange chromatography (mono-Q column) after exchanging the buffer with buffer C (pH 8.0) (20 mM Tris-HCl, 10 mM NaCl). The protein was eluted with linear gradient of NaCl from 10 to 500 mM. The purified protein was stored at −80 °C in a buffer containing 10 mM

Tris-HCl (pH 8.0), 150 mM NaCl and 10% glycerol. Stock concentration of the purified GFP1-10 was 100 μM.

On the other hand, the template DNA for in vitro or in vesicle synthesis of the smaller partner (GFP11) was prepared by PCR using P18 and P19 primers, where P17 oligo was used as the template. The amplified DNA was cloned into pET29a vector using NdeI and EcoRI. The resulting construct was used as a template DNA for the second PCR using T7 promoter and terminator primers (P22 and P23) to prepare the template DNA for GFP11 synthesis in the PURE system (Supplementary Fig. 15). The further PCR for making the template DNA for GFP11 photosynthesis in vesicle (Fig. 2g, h) was carried out using P24 and P20 primers, and the resulting template DNA omits the non-translational sequence after the stoop codon.

**Reconstruction of PLs.** The reconstitution of PLs with either bR or $F_oF_1$ or the co-reconstitution of bR$F_oF_1$-PLs has been performed based on complete detergent solubilization of liposomes following previous literature[10,19,34] by the incorporation of the necessary modifications. Buffer PA6-5 was used as a reconstitution buffer unless otherwise indicated. First, lipid powder was suspended in buffer PA6-5 (pH 7.3), composed of 10 mM HEPES, 3 mM MgCl₂, 10 mM NaH₂PO₄ and 200 mM sucrose, at a concentration of 41.3 mM. Later, the lipid suspension was completely solubilized by 6% octyl β-D-glucopyranoside (W/V) for about 1 h at room temperature. Then, the detergent was removed by first-round addition of 200 mg of pre-equilibrated SM2 Bio-Beads (Bio-Rad) and incubated at room temperature for 30 min. This was followed by the addition of bR (as a purple membrane), $F_oF_1$ or both bR and $F_oF_1$ at a given final concentration. The incubation was continued for additional 30 min, before the second-round addition of 300 mg of Bio-Beads by rotation mixing. After adding the second-round Bio-Beads, the proteoliposome was left mixing at room temperature for 90–120 min. Then, the turbid upper fraction of the proteoliposome was collected and stored at −80 °C until use.

**Light-dependent proton-pump activity of bR-PLs.** For assaying proton pumping activity of bR, 800 μl of 1 R buffer (10 mM HEPES-KOH (pH 7.3) and 3 mM MgCl₂), pre-warmed at 37 °C, was supplemented with 30 μl bR-PL and 0.5 μg ml⁻¹ ACMA. Thereafter, the fluorescence trace of ACMA was measured every 0.2 s at the excitation and emission wavelength of 410 nm and 480 nm, respectively, using JASCO FP6500 spectrofluorometer that is customized by attaching a light source (Tokina, Techno light KTS-150RSV, equipped with 15 V 150 W lamp and 520 nm long-pass filter) at the lid. For the bR synthesized in the PURE system, an assay buffer composed of 10 mM HEPES-KOH (pH 7.5), 5 mM MgCl₂ and 100 mM KCl was used. The proton-pump activity of bR was initiated by illuminating the sample with the light source.

**Ultrapurification of ADP.** The purchased ADP-monopotassium salt powder was first dissolved with 10 mM HEPES-KOH (pH 7.3) at the concentration of 400 μmol, then applied to pre-equilibrated mono-Q column and eluted with linear salt gradient of 0–300 mM NaCl. The peak fraction for ADP was collected and lyophilized (using Taitec VD-800F freeze-dryer), and suspended with 10 mM HEPES-KOH (pH 7.3) at the stock concentration of 80 mM.

**Light-driven ATP synthesis activity of bR$F_oF_1$-PL.** For the ATP synthesis activity assay, 20 μl PL was mixed with 7.3 mM ultrapurified ADP and 10 mM NaN₃, and the total volume to 40 μl with buffer PA6-5 was filled up. After illuminating the PLs for a given time length and light intensity with a halogen lamp, the ATP synthesis was terminated by breaking the PLs with 2.5% trichloroacetic acid (TCA). This was accompanied by neutralizing the mixture with equal volume of buffer N (250 mM Tris-HCl (pH 9.5) and 4 mM EDTA). Then, the mixture was injected into 800 μl of buffer R (pH 8.3) (20 mM tricine, 20 mM succinic acid, 80 mM NaCl and 0.6 mM KOH) which was pre-supplemented with 100 μl of luciferase/luciferin mix (CLSII). The luminescence was measured (using AB-2270 ATTO luminometer) every 1 s for a total of 150 s. The trace of luminescence signal was used to calculate the synthesized amount of ATP based on the standard curve where the luminescence intensity is plotted against known concentrations of ATP.

**ΔpH-dependent ATP synthesis activity of $F_oF_1$-PL.** The ΔpH-dependent ATP synthesis activity of $F_oF_1$-PL was measured by acid-base transition assay[10,24]. SoyPC extract was reconstituted with 1 μM $F_oF_1$ to form $F_oF_1$-PL at the concentration of 16 mg ml⁻¹. This was used for acid-base transition assay. The lumen of 30 μl $F_oF_1$-PL was acidified in acidification buffer (20 mM succinic acid, 0.6 mM KCl, 2.5 mM MgCl₂, 10 mM NaH₂PO₄ (pH 4.5)) containing 286 nM valinomycin, 0.8 mM ultrapure ADP and 0.07 mM AP5A for 5 min at room temperature. The ΔpH-dependent ATP synthesis was assayed by injecting the acidified PL into a basic buffer (20 mM Tricine, 130 mM KOH, 2.5 mM MgCl₂, 10 mM NaH₂PO₄, 7.3 mM ultrapure ADP (pH 8.8)) containing luciferin and luciferase reagents. The generated ATP level was estimated by injecting 0.2 nmol of ATP three times as a standard.

**Preparation of GUVs**. GUVs were prepared as described previously[35,36], with slight modification. GUVs were prepared from a fresh lipid-paraffin mix always. Lipids dissolved in chloroform at a desired stock concentration were mixed together considering the lipid-paraffin mix volume of 500 μl. This mixture was mixed well and flushed with a flow of $N_2$ gas. To enhance the lipid mixing and completely remove the remaining chloroform, the lipid-paraffin mix was heated at 80 °C for 20 min and vigorously vortexed. This was further flushed with a flow of $N_2$ gas and the vials were sealed tightly. The vials containing the lipid mix were then sonicated in warm water bath (55 °C) for 30–60 min.

The lipid mix was let to cool at room temperature. Subsequently, 300 μl of the lipid-paraffin solution was mixed with 30 μl inner solution supplemented with 200 mM of sucrose. Water-in-oil emulsion was created by gently pipetting up and down for few 10 s. The emulsion was slowly laid over the outer solution (composed of equal volume of 400 mM glucose and PURE buffer, which is the same composition as GUV inside but without tRNAs) and kept on ice for 10 min. Next, the sample was centrifuged at $10,000 \times g$ for 30 min followed by collecting the precipitate by piercing the bottom of the flat-bottomed Eppendorf tube by 18-gauge needle. Finally, the collected GUV suspension was centrifuged again at $9100 \times g$ for 10 min and precipitated GUV was resuspended in 50 μl of fresh outer solution.

**Light-driven ATP synthesis inside GUV**. The GUVs were prepared from lipid mixture of POPC, cholesterol and PEG2000PE (5.75:4:0.25 molar ratio). The bRF$_o$F$_1$-PLs was constituted from 176 μM bR and 1 μM F$_o$F$_1$ with buffer-1 (pH 7.3), and then encapsulated inside GUVs together with 7.3 mM ultrapurified ADP and 10 mM $NaN_3$. After the formation of GUVs, the outer buffer was replaced by a fresh buffer-1 containing 200 mM glucose. Proteinase K (PK) was supplemented either to the outer or both the inner and outer solution at the concentration of 1 μM. After PK addition, the samples were incubated at 37 °C for 2 h prior (to degrade unencapsulated bR and F$_o$F$_1$) to the main light or dark incubation. These GUV samples were incubated at 37 °C under light or dark. As a control, an in vitro reaction mixture was also prepared in the same condition as the encapsulated reaction mixture and illuminated.

**Light-driven protein synthesis in GUV or in vitro**. For the light-driven translation reaction, bRF$_o$F$_1$-PL (176 μM bR and 1 μM F$_o$F$_1$) was resuspended in buffer-1 (Supplementary Table 4) supplemented with 500 nM of mRNA encoding sfGFP or bR::sfGFP (see the DNA sequence in Supplementary Table 5), 7.3 mM ADP, 10 mM $NaN_3$, 40 A per ml tRNA and 200 mM sucrose. This was mixed with the corresponding enzyme mix (Supplementary Table 2) and then encapsulated inside GUVs. The vesicle suspension was incubated for 6 h at 37 °C under the light illumination passing 500 nm long-pass filter. For the light-driven transcription-and-translation reaction, the reaction mixture was prepared in same way, but using buffer-2 (Supplementary Table 4) supplemented with 1 nM DNA encoding GFP11 (see the DNA sequence in Supplementary Table 5) and 5 μM of the purified GFP1-10 protein. The reaction was carried out for 7 h at 37 °C under the light illumination.

The vesicles synthesizing sfGFP were observed by confocal laser scanning microscopy (CLSM) (Zeiss LSM 550). For the population analysis, the vesicle suspension was first diluted 10 times with buffer PA6-5 and then the fluorescence intensity of 100,000 vesicles was analyzed by fluorescence-activated cell sorter (FACS Aria III).

In vitro, the light-driven transcription-translation reaction was performed using the same reaction mixture as mentioned above in the presence of [$^{35}$S]methionine. The synthesized protein was subjected to 15% SDS-PAGE and visualized by autoradiography (Fujifilm). For the light-driven transcription-and-translation reaction, the sample was reacted for 13 h and subjected to column filtration (RNase-Free Micro Bio-Spin Columns with Bio-Gel P-30). Radioactivity of the resulting sample was counted by Liquid Scintillation Counter (ALOKA LSC-6100).

Concerning light-driven translation reaction using transient light illumination, 500 μM of ADP, 500 nM mRNA coding GFP and [$^{35}$S]methionine were supplemented in the translation reaction mixture. At a given time, aliquots were taken for 15% SDS-PAGE and ATP amount analysis.

**bR::sfGFP synthesis in GUV**. GUVs were prepared with 6 mM POPC and 4 mM cholesterol with encapsulating a cell-free reaction mix (PURE*frex* Ver.2) together with 5 nM linear DNA of bR::sfGFP construct (see the DNA sequence in Supplementary Table 5), 150 μM ATR and 8 mg ml$^{-1}$ of SoyPC liposome. The reaction was performed in the presence or absence of liposomes for 4 h at 37 °C without light illumination. The fluorescence pattern of the synthesized bR::GFP fusion protein in GUV lumen was analyzed by CLSM.

**Preparation of bR or F$_o$ DNA for the cell-free expression**. The bR gene (with the codon optimized for expression in the PURE system) was amplified by primer P26 and P31, in the first step of PCR, whereas P21 and P31 primers were used in final step of PCR. Furthermore, for bR::sfGFP fusion construct, 3' and 5' overlap sequences were added to the bR and sfGFP gene by separate PCR using a primer set of P25 and P27 or P28 and P29, respectively. The resulting PCR products were used for overlap PCR in the presence of P25 and P29 as a forward and a reverse primer, respectively. Eventually, ribosome binding site and T7 promoter site were added to the fusion construct by the primer P26 and P21, respectively, in the final

two-step PCR (i.e., where P30 was used as a reverse PCR). The DNA templates for the cell-free expression of F$_o$ complex (*uncB*, *uncF* and *uncE*) were prepared using a primer set of P32 and P35, P36 and P37, or P38 and P39, respectively. Point mutation in *uncB* (R169A) was incorporated by overlap PCR using P33 and P34. Finally, T7 promoter, ribosome binding site and 3' extra 14 nucleotides were added to it by P32 and P35.

**Co-flotation assay**. Spontaneous membrane integration of cell-free synthesized bR was performed as described previously[37] with modification. bR was synthesized in the PURE system from 5 nM of bR template DNA (with upstream T7 promoter and ribosome binding site) in the presence of [$^{35}$S]methionine and 8 mg ml$^{-1}$ SoyPC extract liposomes, which was filtered by 200 nm pore size membrane using a micro-extruder (Avanti Polar). The synthesis reaction was terminated by adding RNaseA at the concentration of 20 ng μl$^{-1}$ followed by incubation at 37 °C for 30 min. Then, 10 μl of synthesized bR (i.e., bR-PL) was overlaid on the top of 0, 25 and 30% sucrose layers, where each layer was with a volume of 300 μl in sucrose flotation buffer (50 mM HEPES-KOH (pH 7.6), 100 mM KCl, 10 mM $MgCl_2$). This was centrifuged at $157,800 \times g$ for 30 min using Beckman Coulter Optima Max XP Ultracentrifuge using TLS 55 rotor. Fractions of 200 μl were collected from the top in a total of four fractions. To the collected fractions, ice-chilled TCA was added at the concentration of 10% and centrifuged at $20,400 \times g$ for 30 min at 4 °C. The supernatant was removed and the pellet was resuspended in 200 μl of acetone and sonicated for at least 3 min. The suspension was centrifuged again at $20,400 \times g$ for 30 min. The pellet was left at room temperature for 5 min, mixed with 10–20 μl of loading dye and water-bath sonicated for 1 min. The sample was finally analyzed by 15% SDS-PAGE.

**Membrane orientation assay**. Membrane orientation of the reconstructed bR-PLs or F$_o$F$_1$-PLs was assessed by observing the binding affinity of each protein component to Ni-NTA Magnetic Beads, and 41.3 mM of lipid mixture (SoyPC extract/cholesterol with the molar ratio of 70:30) was reconstituted with either F$_o$F$_1$ or bR at the protein concentration of 0.2 μM. The β-subunit of the recombinant F$_o$F$_1$ and C-terminus of bR were bearing a His-tag. Magnetic beads conjugated with Ni were used to trap His-tagged terminus of the proteins facing outside of liposome (cytosol side). As a control, PLs were solubilized with 0.5% Triton X-100 and had to undergo the same procedure as the experimental samples. Finally, the collected fractions were run in 10–20% gradient gel and bands were visualized by western blotting with anti-His-tag antibody.

**Photosynthesis of bR or F$_o$**. A mutant bR (bR$_{mut}$) containing substitutions of D85N and K216N was used as a negative control of photosynthesis of bR. The point mutations were introduced into pEXP-5-ct-bRwt construct, sequentially, by quick change PCR using primer sets of P32-and-P33 and P34-and-P35 (see the DNA sequence in Supplementary Table 5). Next, linear DNA template was prepared using P22 and P23 to be used as a template for in vitro transcription.

For light-dependent expression of either bR$_{mut}$ or functional bR$_{wt}$, the reaction mixture of PURE system was supplemented with bRF$_o$F$_1$-PL (where the PL was reconstituted with 5 μM bR and 1 μM F$_o$F$_1$), 100 μM of ATR and either 0.4 μM of bR$_{wt}$ mRNA or 0.8 μM bR$_{mut}$ mRNA. The reaction mix was incubated at 37 °C under light and 2 μl sample was aliquoted at each given time for subsequent ATP level quantification.

On the other hand, for functional in vitro assembly of F$_o$F$_1$, the F$_o$ complex was expressed from the *uncB*, *uncF* and *uncE*-mRNAs (i.e., at the concentration of 40 nM, 20 nM and 100 nM, respectively) coding for their respective subunits *a*, *b* and *c*. The expression of the F$_o$ complex was supplemented with 300 nM of purified F$_1$ complex for co-translational assembly of the two partners. The reaction mixture of PURE system (buffer-2 containing 200 mM of sucrose) was initially seeded with 18 nM of bRF$_o$F$_1$-PL that was composed of 140 μM bR and 0.29 μM F$_o$F$_1$. After incubating the reaction mixture for 7 h under light at 37 °C, 5 times diluted PL was isolated by centrifuging at $230,000 \times g$ for 30 min. The precipitated PL was resuspended in assay buffer (pH 7.3) (composed of 10 mM HEPES, 7.3 mM ADP, 10 mM $NaH_2PO_4$, 3 mM $MgCl_2$, 10 mM $NaN_3$ and 200 mM sucrose), making the final concentration of 44 nM. Later, the light-driven ATP synthesis was assayed as described before. For negative control, the mutant *uncB* (R169A) mRNA template was used.

**Reporting Summary**. Further information on experimental design is available in the Nature Research Reporting Summary linked to this article.

## Data availability

The authors declare that all the relevant data supporting the findings of the study are available in this article and its Supplementary Information file, or from the corresponding author (Y.K.) upon reasonable request.

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

## Acknowledgements

We thank Professor D. Oesterhelt and Dr. S.V. Gronau for the *Halobacterium salinarum* strain R1, Dr. Kazuhito Tabata for discussion on ATP synthase, Dr. Satoshi Fujii for operation of FACS, Dr. Toshiharu Suzuki for the constructs of $F_oF_1$ and Dr. T.Z. Jia for advice on manuscript preparation. This work was supported by JSPS KAKENHI (Grant Numbers 16H06156, 16KK0161, 16H00797, J26106003 to Y.K. and 16H02089 to T.U.), the Astrobiology Center Project of the National Institutes of Natural Sciences (NINS) (Grant Number AB291017 to Y.K.), and JST, PRESTO (Grant Number JPMJPR18K5 to Y.K.).

## Author contributions

S.B. performed all experiments. Y.K. conceived the idea, designed most of the experiments, analyzed the data, prepared the figures and wrote the manuscript. T.U. gave advice for manuscript preparation.

## Additional information

**Competing interests:** The authors declare no competing interests.

