## [Peer Review File · Nature Communications]

Reviewers' Comments:

Reviewer #1:

Remarks to the Author:

In this manuscript by Berhanu et al., the authors prepared an artificial cell containing bacteriorhodopsin (bR), a light-driven proton pump, and the F1F0 ATP synthase, and utilized the organelles for autotrophic photosynthesis. To prove the functional ability of the artificial cell, the authors provide experimental data on light-driven proton consumption and ATP production, GFP synthesis using the produced ATP, and energetically-independent photosynthetic system by replacing the target protein (GFP) with bR in the artificial system. The manuscript is well written and the experiments well designed. Moreover, I believe that this manuscript describes an important topic area that is growing in impact. In recent years, for example, there are many emerging activities focused on the ability to build a cell from the ground up. That said, while potentially exciting, the manuscript felt like a natural extension of previous work and I didn't find any of the results particularly surprising or novel on their own. For example, I would expect that ATP generated could be used for transcription and translation if the right machinery and energy nucleotide regeneration systems are in place. In addition, the concept of using bR in artificial cells has been historically pursued (as the authors themselves highlight), and I didn't quite understand why the authors got this to work where others may have failed. Given the system inefficiencies (it seems like anywhere from 15-50% of the GUVs had the desired activity if I understand correctly), the actual impact is unclear, especially given conventional approaches to produce ATP in liposome based artificial cells have been reported at higher levels. I also found some of the commentary in the discussion a bit speculative, focusing more on hypothesizing the emergence of primordial cells than on the data. These features lessened my enthusiasm and made me feel like Nature Communications might not be the right match for publishing the work. Below I describe some of my major concerns of the work.

Major concerns

1. If I understand correctly, the F1F0 ATP synthase is in all cases purified and added to the reaction. This means that only one organelle is actually made by the cell-free system in the third part of the paper. Is this correct? Have you tried also to make the F1F0 ATP synthase? This has been reported (<https://www.ncbi.nlm.nih.gov/pubmed/21925509>) and doing so here would elevate the impact.
2. As I understand, previous cell-free systems have used bR <https://www.ncbi.nlm.nih.gov/pmc/articles/PMC2786979/>. What was special here that allowed for the ATP to be regenerated at a higher level?
3. In the light driven protein synthesis section, do I understand correctly that only ~50% of the GUVs made GFP. Why not all of the GUVs? This is important to resolve as it would be critical to truly building an artificial cell? The authors should identify and alleviate the limiting component(s), which might also help their efforts to build an energetically recursive system.
4. The autotrophic artificial cell described in this manuscript is based on light-driven ATP synthesis in which the authors design the cell to produce bR which facilitates the production of ATP. Although I agree the design of the artificial cell for self-sustaining protein synthesis is a potential strong point in the manuscript, I am concerned about the observation of the 1.5-fold increase (line 201) in the wt versus mutant bR and whether or not this is significant. Is this significant? Is there any chance that the wt version enhances bR activity of the 5uM seeded components?
5. With respect to the self-sustaining system, the authors should describe information on previous examples producing ATP to demonstrate significance. For example, how does this compare to the 2018 Nature Biotechnology paper they refer to by Lee et al.?
6. What is the maximum reaction duration with the system? The 2018 Nature Biotechnology paper from Lee et al. showed ATP conversion for 3 days (half-maximum efficiency) at room temperature and 1 month at 4C. The maximum observation time shown in this manuscript is only 5 h, which is low compared to the previous result. Additional discussion on the sustainability of the authors' system would improve the quality of the manuscript, but as it stands it seems to fall short of past

works. Can the authors extend their reactions to a day for example?

Minor concerns

- Line 40: The yield represented in concentration (1.8 mM) may mislead the ATP production capability of the system given the constrained volume. Please provide the number of ATP molecules produced for direct comparison with other previous literature.
- Fig. 1A shows that bR captures light energy and uses it to move protons across the membrane and the authors use 'the decrease of ΔpH ' as a proof that protons are pumped into the cell. Please depict in Fig. 1C or describe in the main text how pH of the outside of the cell is measured. Although the authors did not define ΔpH , I assume $\Delta\text{pH} = \text{pH}(\text{original, out}) - \text{pH}(\text{current, out})$. If so, when the concentration of protons outside decreases (when bR starts pumping H^+ into the inside), $\text{pH}(\text{current, out})$ increases, which makes ΔpH larger. Therefore, line 49 should be described as: 'increase of ΔpH caused by bR' or 'increase of pH outside caused by bR', if the authors are measuring pH of the outside. I apologize for my confusion.
- Line 67: Lacking the strain of bacterial cells (*Halobacterium salinarum*?).
- Fig. S4. Identify which bands is the target protein. It clearly shows two bands. Please label all bands.
- Line 79, Describe how the orientation and net-working ratio are 'normalized'. For example, what did normalization do to increase the first value of $\sim 78.4\% \rightarrow \sim 86\%$. Also, please use the same significant figures throughout the manuscript.
- Please provide a standard deviation for the average liposome size. Fig. 2A shows a variation of the sizes, but there is no discussion in the manuscript how the size of GUVs is determined. The size of GUV is important because it determines the volume of the cell, which determines the concentration of produced ATP.
- Line 96, should the amount of produced ATPs be 0.6×10^6 here and not 6×10^6 . See Fig. 1E, The y-axis in from 0 to 1 ($\times 10^6$).
- Line 99, please explain how NaN_3 inhibits the reverse activity of ATPase or refer to previous literature. Have other inhibitors been tested that impact the ATPase or the membrane potential?
- Line 105, please discuss the number of produced ATPs rather than concentration for comparison to other literature.
- I calculated the turnover of ATP production in a single GUV using the values provided in Table S1. Since the authors obtained 1.8 mM (which they claim 50,962 ATPs) in 6 h, the turnover is $50,962 / 6 \text{ h} = 2.3 \text{ s}^{-1}$, which is 20-fold lower than the turnover obtained in a single PL. Did I do this correctly? The authors should discuss the difference in turnover between PLs and GUVs in relation to this and other works.
- The title of the table is 'GUV vs. bulk', but they say PL in the table. Which is correct?

Reviewer #2:

Remarks to the Author:

Building artificial cells using cell-free protein synthesis (CFPS) has become a widespread exercise. Many laboratories are working on this type of research. The idea is to engineer life from scratch using molecules, with goals including both applications and fundamental research. CFPS is a preferred technique because it permits gene expression *in vitro*. In this work, entitled "Autotrophic artificial minimal cell recursively producing energy for protein synthesis", the authors purify a light-driven ATP regeneration system composed of the bacteriorhodopsin and the ATP synthase and incorporate the proteins into the membrane of lipid vesicles. They show that this system regenerates ATP. They use this system with the PURE CFPS system to show that the regenerated ATP can be used for the translation of proteins (a fluorescent reporter protein and the bacteriorhodopsin). The work related to ATP regeneration is well done and well described. This experiment reproduces the work done by Lee et al in 2017 ("Photosynthetic artificial organelles sustain and control ATP-dependent reactions in a protocellular system"). The work done with the PURE system is certainly a good step towards artificial cells capable of regenerating ATP using light. The way the experiments are presented, however, is overstated and exaggerated with

respect to what the authors really show. It's a first step towards an artificial cell using CFPS working with a photosynthetic system, but this artificial cell system is far from being autotrophic and self-sustaining.

Major concerns:

- prior publications on a similar topic are not well acknowledged. For example, the work by Lee et al. in 2017 ("Photosynthetic artificial organelles sustain and control ATP-dependent reactions in a protocellular system") is cited but similarities to many results presented in this work are not discussed. The work shown in figure 1 is a repetition of the work reported in the paper by Lee and coworkers.
- Some claims are overstated, especially on the fact that the artificial cell is self-sustaining and autotrophic. It is a very strong claim. Unfortunately, the results presented in this work do not demonstrate that the artificial cell is self-sustaining or autotrophic. The authors do not show that the strength of the light-driven ATP regeneration is large enough to sustain the de novo expression and synthesis of the whole set of enzymes (bacteriorhodopsin and ATP synthase) and efficient expression of other genes such as eGFP. By definition, an autotrophic organism is capable of self-nourishment by using inorganic materials as a source of nutrients and using photosynthesis or chemosynthesis as a source of energy. The light-driven ATP regeneration seems far to be enough to permit self-maintenance and autotrophy.

Other comments:

- (a) The title claims too much with respect to the results. Autotrophy is not achieved and recursively is also too strong. The title should mention that it's about CFPS.
- (b) Abstract: (1) Line 13: Attempts to construct an, (2) Line 19: The artificial cell contains purified photosynthetic, (3) Line 25: cell produced chemical energy and
- (c) Introduction: (1) Line 29: let to whole cell reconstruction: this was never done really, (2) Line 38: In this study, we have studied that how to apply the: it's hard to understand, (3) Lines 40-44: the last sentence of the intro is way too strong: self-sustaining artificial cell and resulting in an energy-independent feedback loop, these things are not demonstrated. If it would be the case, we would see a continuous production of large amounts of proteins, over many days. The level of protein expressed here is incredibly small.
- (d) Results: (1) Line 76: Thus to inhibit the fluidity What about: Thus to decrease the leak through the membrane, (2) Line 121: significant fluorescence: give numbers, like this it does not mean anything. (3) Figure 2F: what is the ratio between the two bands on the gel? (4) Line 188: and likely contributed: the authors do not seem sure of their claim. (4) Figure 3: it's not a recursive system. It would be recursive if the light-driven regeneration provides the energy to synthesize the whole system (bacteriorhodopsin and ATP synthase, plus some other proteins like a reporter).
- (e) Discussion: (1) Line 235: The chemical energy in ATP is a fundamental, (2) Line 242: translated into a functional protein, GFP: not really, what's produced is only a small piece of GFP. (3) Line 249: It should be noted that all, (4) Line 265: Our work demonstrated that, this work was already demonstrated by Lee et al in 2017.
- (f) other comments: a more convincing experiment would have been to synthesize the bacteriorhodopsin and ATP synthase using the ATP regeneration system added to the PURE (kinase and phosphate donor) and demonstrate that the synthesized light-driven system can take over the ATP regeneration to extend production of a reporter gene for days based on light only.

The supplementary information is composed of 4 tables, 17 figures, and the methods. Some comments:

- (a) line 34: were (and not was).
- (b) line 39: Aldrich.
- (c) line 40: 45,000 g (same on line 45).
- (d) line 41: more than 6 times, how many times really?
- (e) line 49 and others: give complete coding sequence as text in the supplementary material for each gene cloned in this work, including promoter sequence.

- (f) line 54: do you mean French press?
- (g) for all the protein purified, specified the concentration of the stock solutions.
- (h) line 123: as deemed necessary, that's not a way to report scientific work, be more clear.
- (i) line 170: 16 mg/ml
- (j) line 191: PURE buffer, which one?
- (k) line 197: were (not was).
- (l) line 220: 100,000
- (m) spaces
- (n) line 231: 8 mg/ml
- (o) line 258: 157,800 g
- (p) fig. S1: specify the kDa of bR.
- (q) fig. S2: what are the units for the slopes in the table? Same for fig. S3.
- (r) fig. S7: specify the concentration of bR and FOF1 used.
- (s) fig. S13: why are the signals for column 5 and 7 that high? They should be at background level.
- (t) fig. S14: error bar, do you mean scale bar? What does 'Non' mean?

Reviewer #3:

Remarks to the Author:

The manuscript by Berhanu and colleagues describes construction of a synthetic minimal cell that can generate energy to sustain protein synthesis.

This is well designed and technically very sound paper.

This paper is touching on very similar subject to the recently published work: Lee, K. Y. et al. Photosynthetic artificial organelles sustain and control ATP-dependent reactions in a protocellular system. *Nat. Biotechnol.* (2018). doi:10.1038/nbt.4140

In that Nature Biotechnology paper, authors reconstituted ATP synthesis machinery embedded in a membrane of artificial synthetic cell organelle.

The present manuscript by Berhanu and colleagues is based on a very similar concept.

Minor points:

figure 1a is rather unclear. I get that the authors were aiming mostly for visual appeal here, but it's hard to understand what is going on.

The lipids used for giant unilamellar vesicle formation include both POPC and soyPC, which for the purpose of membrane formation are nearly the same thing. It would be interesting to note why authors decided to use those lipids, and also how was the composition of membrane decided. As in case of most membrane proteins, the specific composition and size of membranes has usually big impact on measured activity.

In encapsulation experiments, there is usually large amount of liposomes that do not express genes from the encapsulated plasmids. Did authors ever investigate what percentage of "dark" liposomes observed in flow cytometry were "dark" because they did not encapsulate all necessary components, vs had all components but the circuit did not work for some reason? This would be important distinction if this technology is to be used for construction of more complex synthetic minimal cell.

What is the overall yield, or efficiency, of ATP synthesis? comparison to natural photosynthesis would be very useful for further development of this system.

Reviewer #1 (Remarks to the Author):

In this manuscript by Berhanu et al., the authors prepared an artificial cell containing bacteriorhodopsin (bR), a light-driven proton pump, and the F₁F₀ ATP synthase, and utilized the organelles for autotrophic photosynthesis. To prove the functional ability of the artificial cell, the authors provide experimental data on light-driven proton consumption and ATP production, GFP synthesis using the produced ATP, and energetically-independent photosynthetic system by replacing the target protein (GFP) with bR in the artificial system. The manuscript is well written and the experiments well designed. Moreover, I believe that this manuscript describes an important topic area that is growing in impact. In recent years, for example, there are many emerging activities focused on the ability to build a cell from the ground up. That said, while potentially exciting, the manuscript felt like a natural extension of previous work and I didn't find any of the results particularly surprising or novel on their own.

We do not think our work is just a natural extension of Lee's previous work because we demonstrated protein synthesis using photogenerated ATP. Additionally, we also showed that so synthesized bR and F_o portion of ATP synthase became a part of the artificial organelle and enhanced the ATP photosynthesis activity. Previous works have not succeeded in the synthesis of protein using light energy. In our artificial cell system, we could synthesize any protein just by changing input DNA. This is important when we think of a self-reproducible artificial cell that will be continuously alive after the self-division. We would like to appeal this point strongly to the reviewer#1.

For example, I would expect that ATP generated could be used for transcription and translation if the right machinery and energy nucleotide regeneration systems are in place.

It is true that the ATP produced by the right machinery and energy regeneration system could be used for transcription and translation. The point we advocate is the product of the translation do feedback to the machinery. This is conceptually new point comparing the previous reports.

In addition, the concept of using bR in artificial cells has been historically pursued (as the author themselves highlight), and I didn't quite understand why the authors got this to work where others may have failed.

We could get this bR-F_oF₁-coupling system to work in the artificial cell system because we used sodium azide that inhibits the reverse activity (ATP-dependent H⁺-pump activity) of ATP synthase. Additionally, we optimized the preparation method of bR-proteoliposome to improve the ratio of the right membrane oriented bR. We added some sentences explaining this point in the main text, Supplementary information (Fig. S3), and the section of materials and methods. Moreover, the reconstituted cell free system has allowed us to manipulate the components so as to construct the desired artificial photosynthetic cell that is only relayed on photosynthesized ATP as an energy form.

Given the system inefficiencies (it seems like anywhere from 15-50% of the GUVs had the desired activity if I understand correctly), the actual impact is unclear, especially given conventional approaches to produce ATP in liposome based artificial cells have been reported at higher levels.

The ratio which mentioned by the reviewer#1 is that of protein (GFP) synthesizing vesicles, not ATP synthesizing vesicles. The reason why the ratio does not show 100% is that the vesicle needs to encapsulate enough amount of all components of the PURE system and the DNA encoding *gfp* to synthesize GFP inside and the bRF_oF₁-PLs.

I also found some of the commentary in the discussion a bit speculative, focusing more on hypothesizing the emergence of primordial cells than on the data.

We removed the suggested sentence and rewrote as follows.

“Thus, we think that primordial cells using sunlight as a primal energy source could have existed in the early stage of life before evolving into an autotrophic modern cell system.”

These features lessened my enthusiasm and made me feel like Nature Communications might not be the right match for publishing the work. Below I describe some of my major concerns of the work.

We appreciate the reviewer#1 for the comment. The followings are our response for the points arisen by reviewer#1.

Major concerns

1. If I understand correctly, the F1F0 ATP synthase is in all cases purified and added to the reaction. This means that only one organelle is actually made by the cell-free system in the third part of the paper. Is this correct? Have you tried also to make the F1F0 ATP synthase? This has been reported (<https://www.ncbi.nlm.nih.gov/pubmed/21925509>) and doing so here would elevate the impact.

As the reviewer#1 suggested, we only used the purified F₀F₁ ATP synthase in this study. The reason why we did not try to synthesize F₀F₁ is that F₀F₁ is composed of 8 subunit proteins whereas bR has only one. We can synthesize the 8 kinds of protein in the PURE system, but to regulate the stoichiometry of each product is difficult. It is true that the indicated paper by Matthies *et al.* synthesized the proteins in a cell-free system, but that the cell-free system is an extract-base system. Our cell-free system is, on the other hand, a reconstructed system made of each purified molecule for translation, thus the

protein synthesis activity is not as high as the extract-based one. We do not use the extract-based system because we cannot control all the components within the extract. This will be a problem when we develop “self-reproduction” in artificial cell.

In fact, we have tried to synthesize F_0F_1 from 8 kinds genes in the PURE system, but we could not observe significant activity of the product due to the less protein productivity. However, we could detect the F_0F_1 activity when we synthesized only F_0 part (the membrane embedding portion) in the presence of purified F_1 . We also detect the light-driven ATP synthesis activity in co-working with bR. Based on this result, we newly tried to photosynthesize F_0 component proteins which composing functional wildtype a or inactive mutant a -subunit, as with the case of bR photosynthesis. This try resulted in that the ATP synthesis rate of artificial organelle became faster when the wildtype a -subunit was photosynthesized together with b and c -subunit proteins. This means the ability of bR F_0F_1 -PLs was strengthened by positive feedback of the *de novo* photosynthesized F_0F_1 . Although our new results have not completely satisfied the reviewer#1's suggestion, we could show that our artificial cell system is working as designed even for F_0F_1 and could elevate the impact of our research.

2. As I understand, previous cell-free systems have used bR <https://www.ncbi.nlm.nih.gov/pmc/articles/PMC2786979/>. What was special here that allowed for the ATP to be regenerated at a higher level?

The paper suggested by the reviewer#1 shows the proton pump activity of cell-free synthesized bR. Our system also synthesized bR, although the artificial organelle (bR F_0F_1 -PLs) is composed of the purified bR. The main difference in our system is that no detergent was contained in the cell-free system, therefore the bR protein integrated into the liposome membrane co-translationally. This happened by a hydrophobic interaction of lipid membrane and hydrophobic

property of the membrane protein. Other difference is that our system contains NaN₃ during the reaction of ATP photosynthesis in order to inhibit reverse activity of ATP synthase (ATP-dependent H⁺-pump activity). This helps the efficient cooperative function of bR and F_oF₁.

3. In the light driven protein synthesis section, do I understand correctly that only ~50% of the GUVs made GFP. Why not all of the GUVs? This is important to resolve as it would be critical to truly building an artificial cell? The authors should identify and alleviate the limiting component(s), which might also help their efforts to build an energetically recursive system.

In order to make a successful artificial cell, all of the PURE components must be encapsulated in a vesicle in proper concentrations. The PURE system consists of 36 kinds of enzymes, ribosome, 40 kinds tRNAs, template DNA (or mRNA), and small molecular compounds. To encapsulate these all molecules in each vesicle as same ratio is statistically implausible. Therefore, protein synthesis inside the microcompartments is highly diverse in terms of rate and amount of synthesized protein. Additionally, a recent paper reports some extrinsic factors (solute partition) affect to gene expression in compartments (Synthetic Biology, 2018, 3(1): ysy011). As the reviewer#1 suggested, we will try to solve this problem to build the efficient energetically recursive system

4. The autotrophic artificial cell described in this manuscript is based on light-driven ATP synthesis in which the authors design the cell to produce bR which facilitates the production of ATP. Although I agree the design of the artificial cell for self-sustaining protein synthesis is a potential strong point in the manuscript, I am concerned about the observation of the 1.5-fold increase (line 201) in the wt versus mutant bR and whether or not this is significant. Is this significant? Is there

any chance that the wt version enhances bR activity of the 5uM seeded components?

The number, 1.5-fold, itself may not be a significant as the reviewer#1 suggested. However, we detected the similar enhancement in all three independent measurements. This result was newly added in Fig. S20. In order to enhance more, we have to synthesize more bR. This might be restricted by the ability of PURE system or ATP photosynthesis activity of bRF_oF₁-PL which containing less amount of bR.

5. With respect to the self-sustaining system, the authors should describe information on previous examples producing ATP to demonstrate significance. For example, how does this compare to the 2018 Nature Biotechnology paper they refer to by Lee et al.?

The *Nature Biotechnology* paper by Lee et al. was using similar principle in constructing the artificial photosynthetic organelle where they have reconstituted photosystem II and ATP synthase as the machinery of artificial organelle. They used the produced ATP for the reaction of carbon fixation or actin polymerization. In terms of energy flux, the energy taken from the external environment was just consumed unidirectionally in their artificial cell system. On the other hand, in our system, we used the produced ATP for the expression of functional bR. This means that the obtained energy was consumed for an autogenous growth within the scheme of positive feedback loop. Additionally, we have also succeeded to synthesize F_o portion of ATP synthase. This means that, in theory, our system can continuously generate the own components even when the artificial cell did self-division. These are clearly different points from the previous studies. This issue was additionally described in the introduction section.

6. What is the maximum reaction duration with the system? The 2018 Nature Biotechnology paper from Lee et al. showed ATP conversion for 3 days (half-maximum efficiency) at room temperature and 1

month at 4°C. The maximum observation time shown in this manuscript is only 5 h, which is low compared to the previous result. Additional discussion on the sustainability of the authors' system would improve the quality of the manuscript, but as it stands it seems to fall short of past works. Can the authors extend their reactions to a day for example?

The time referred by the reviewer#1 is the time stably preserving artificial organelle that can maintain the photosynthetic activity. In fact, they measured the photosynthesis reaction within 15 min (Fig. 2h, see right), which is much lower than our data. Thus, the activity of our system is not low as compared to the previous result.

Minor concerns

- Line 40: The yield represented in concentration (1.8 mM) may mislead the ATP production capability of the system given the constrained volume. Please provide the number of ATP molecules produced for direct comparison with other previous literature.

In the previous literature by Montemagno et al. in 2005 (Nano Letters), they described as “ATP production (nmol/mg ATP synthase)”. Because they used exactly the same proteins as ours, bR from *H. salinarum* and FoF1 from *Bacillus PS3*, we followed their style. We added the sentence of “, where 4.6 $\mu\text{mol ATP/mg ATP synthase}$ was produced after illuminating” in the introduction section.

- Fig. 1A shows that bR captures light energy and uses it to move protons across the membrane and the authors use ‘the decrease of ΔpH ’ as a proof that protons are pumped into the cell. Please depict in Fig 1C or describe in the main text how pH of the outside of the cell is

measured. Although the authors did not define ΔpH , I assume $\Delta\text{pH} = \text{pH (original, out)} - \text{pH (current, out)}$.

We apology for the confusion about this thing. As the reviewer#1 pointed, we showed the difference between $\text{pH (original, outside)} - \text{pH (after illumination, outside)}$ in this graph. In order to avoid the confusion, we rewrote the Y-axis of Fig. 1C as " ΔpH at the bR-PLs exterior". And we added the following sentence in the Fig. 1C legend, "Proton pump activity of bR was measured by monitoring the proton concentration at the outside of bR-PLs where fluorescent proton sensor ACMA was added. We defined as $\Delta\text{pH} = \text{pH (original, outside)} - \text{pH (after illumination, outside)}$."

If so, when the concentration of protons outside decreases (when bR starts pumping H^+ into the inside), pH (current, out) increases, which makes ΔpH larger. Therefore, line49 should be described as: 'increase of ΔpH caused by bR' or 'increase of pH outside caused by bR', if the authors are measuring pH of the outside. I apologize for my confusion.

We eliminated the suggested sentence line 49 and substituted as above.

- Line 67: Lacking the strain of bacterial cells (Halobacterium salinarum?).

We added the name of strain of bR.

- Fig. S4. Identify which bands is the target protein. It clearly shows two bands. Please label all bands.

It has been known that the bR band occasionally appear as multi-bands on SDS-PAGE in several previous literatures (*JBC* 1989 Miercke et al., *JBC* 1984 Sehra et al., *Royal Society of Chem.* 2014 Dutta et al.). In order to make this clear, we added as "The appeared bands are both bR [Miercke, 1989 #52]." in the legend of Fig. S6.

- Line 79, Describe how the orientation and net-working ratio are

'normalized'. For example, what did normalization do to increase the first value of ~78.4 % → ~86 %. Also, please use the same significant figures throughout the manuscript.

We added the sentences explaining the way of the normalization in the main text.

- Please provide a standard deviation for the average liposome size. Fig. 2A shows a variation of the sizes, but there is no discussion in the manuscript how the size of GUVs is determined. The size of GUV is important because it determines the volume of the cell, which determines the concentration of produced ATP.

We added one supplement figure showing the average liposome size and standard deviation in Fig. S12 and a sentence "A large majority of the GUV population was in the range of 10-20 μm as diameter (n=200) (Fig. S12)" in the main text.

- Line 96, should the amount of produced ATPs be 0.6×10^6 here and not 6×10^6 . See Fig. 1E, The y-axis in from 0 to 1 ($\times 10^6$).

Corrected.

- Line 99, please explain how NaN3 inhibits the reverse activity of ATPase or refer to previous literature. Have other inhibitors been tested that impact the ATPase or the membrane potential?

We added a citation (Bald et al. 1998 *JBC*) at this sentence.

- Line 105, please discuss the number of produced ATPs rather than concentration for comparison to other literature.

We added the sentence as "This represents 4.6 μmol ATP were produced per mg ATP synthase."

- I calculated the turnover of ATP production in a single GUV using the values provided in Table S1. Since the authors obtained 1.8 mM (which

they claim 50,962 ATPs) in 6 h, the turnover is $50,962 / 6 \text{ h} = 2.3 \text{ s}^{-1}$, which is 20-fold lower than the turnover obtained in a single PL. Did I do this correctly? The authors should discuss the difference in turnover between PLs and GUVs in relation to this and other works.

The reviewer#1 is misunderstanding Table S1. In the Table S1, we described the number of produced ATP in a GUV. This GUV (radius 5 μm) encapsulated 11244 bRFoF1-PLs (11244 artificial organelles). And, one organelle produced 50962 ATP after 6 hours reaction. We do not think there is much sense to calculate the turnover number per a single PL, but it is important to calculate the turnover of FoF1. We calculated the turnover of the reconstructed FoF1 based on the data of Fig.1E. It resulted in $8.3 \pm 0.3 \text{ s}^{-1}$ that is comparable to the previous report ($4.3 \pm 0.1 \text{ s}^{-1}$) by Lee et al (*Nature Biotech.* 2018). This was described in the main text as “The maximum turnover number for ATP synthesis in initial five minutes was 8.3 ± 0.3 , in the case of $176 \mu\text{M}$ bR/ $1 \mu\text{M}$ F_oF_1 . This was almost double compared to the previous report {Lee, 2018 #41}.”.

We do not understand the mean of the reviewer#1's question “*the difference in turnover between PLs and GUVs*”.

- The title of the table is ‘GUV vs. bulk’, but they say PL in the table. Which is correct?

PL means artificial organelle. GUV means the outer envelope of artificial cell including PURE system and a number of PL. Bulk means the reaction in the PURE system (not encapsulated in GUV). We added “*Artificial organelle consists of bRF_oF₁-PL ($176 \mu\text{M}$ bR: $1 \mu\text{M}$ F_oF₁)” as the asterisk of PL.

Reviewer #2 (Remarks to the Author):

Building artificial cells using cell-free protein synthesis (CFPS) has become a widespread exercise. Many laboratories are working on this

type or research. The idea is to engineer life from scratch using molecules, with goals including both applications and fundamental research. CFPS is a preferred technique because it permits gene expression in vitro. In this work, entitled “Autotrophic artificial minimal cell recursively producing energy for protein synthesis”, the authors purify a light-driven ATP regeneration system composed of the bacteriorhodopsin and the ATP synthase and incorporate the proteins into the membrane of lipid vesicles. They show that this system regenerates ATP. They use this system with the PURE CFPS system to show that the regenerated ATP can be used for the translation of proteins (a fluorescent reporter protein and the bacteriorhodopsin). The work related to ATP regeneration is well done and well described. This experiment reproduces the work done by Lee et al in 2017 (“Photosynthetic artificial organelles sustain and control ATP-dependent reactions in a protocellular system”). The work done with the PURE system is certainly a good step towards artificial cells capable of regenerating ATP using light. The way the experiments are presented, however, is overstated and exaggerated with respect to what the authors really show. It’s a first step towards an artificial cell using CFPS working with a photosynthetic system, but this artificial cell system is far from being autotrophic and self-sustaining.

We thank the reviewer#2 for the kind comments. According to the reviewer#2’s suggestion, we changed some exaggerating descriptions to more moderate explanations throughout the manuscript.

Major concerns:

- prior publications on a similar topic are not well acknowledged. For example, the work by Lee et al. in 2017 (“Photosynthetic artificial organelles sustain and control ATP-dependent reactions in a protocellular system”) is cited but similarities to many results presented in this work are not discussed. The work shown in figure 1 is a repetition of the work reported in the paper by Lee and coworkers.

We added the sentences mentioning about the work by Lee *et al.* and explained the difference from their work in the Introduction section. Additionally, we showed the turnover number for ATP synthesis that is calculated based on the data of Fig. 1E and Table S1, and compared with that of Lee's work, in the Result section.

- Some claims are overstated, especially on the fact that the artificial cell is self-sustaining and autotrophic. It is a very strong claim.

We eliminate the word "Autotrophic" from the title and main text. Along with this, we changed the title as "Artificial photosynthetic cell producing energy for protein synthesis".

Unfortunately, the results presented in this work do not demonstrate that the artificial cell is self-sustaining or autotrophic. The authors do not show that the strength of the light-driven ATP regeneration is large enough to sustain the de novo expression and synthesis of the whole set of enzymes (bacteriorhodopsin and ATP synthase) and efficient expression of other genes such as eGFP.

As the reviewer#2 suggested we have not synthesized whole set of the component enzymes. This has been pointed out also by the reviewer#1 (Major point1) and the Editor. So, we tried to demonstrate photosynthesis of F_oF_1 . Although we could not find the activity of whole synthesis of F_oF_1 due to the low productivity, we could synthesize functional F_o complex (a membrane part of F_oF_1) by light. Additionally, we measured the enhanced ATP synthesis activity of the resulting artificial organelle by photosynthesizing F_o , as same as bR case (Fig.3E). The reason of the low productivity may be due to the limitation of the reconstructed cell-free system (PURE system), not the capability of artificial organelle. By this new result, we could show our artificial cell system is able to produce not only bR but also a part of F_oF_1 . Therefore, we believe that our new result is partly satisfying the matter pointed out by the reviewers and Editor, and increased the significant of this research.

By definition, an autotrophic organism is capable of self-nourishment by using inorganic materials as a source of nutrients and using photosynthesis or chemosynthesis as a source of energy. The light-driven ATP regeneration seems far to be enough to permit self-maintenance and autotrophy.

We eliminate the word of “autotrophic” and “self-sustaining” from the manuscript.

Other comments:

(a) The title claims too much with respect to the results. Autotrophy is not achieved and recursively is also too strong. The title should mention that it's about CFPS.

We changed the title as “Artificial photosynthetic cell producing energy for protein synthesis”.

(b) Abstract: (1) Line 13: Attempts to construct an ……., (2) Line 19: The artificial cell contains purified photosynthetic …., (3) Line 25: cell produced chemical energy and ….

Corrected.

(c) Introduction: (1) Line 29: let to whole cell reconstruction: this was never done really,

We change the sentence as “Recent advances in synthetic biology allow us to challenge whole reconstruction of cell from simple non-living molecules and redesigned minimal genome”.

(2) Line 38: In this study, we have studied that how to apply the ….: it's hard to understand,

We change the sentences as “In this study, we performed ATP synthesis by light-driven artificial organelle inside GUV.”.

(3) Lines 40-44: the last sentence of the intro is way too strong: self-sustaining artificial cell and resulting in an energy-independent feedback loop, these things are not demonstrated. If it would be the

case, we would see a continuous production of large amounts of proteins, over many days. The level of protein expressed here is incredibly small.

We eliminated the word “self-sustaining” and changed the sentences as “By combining the artificial organelle and PURE system, we design and construct an artificial photosynthetic cell that produces ATP for the internal protein synthesis. The produced ATP was consumed as a substrate of mRNA, or as an energy for aminoacylation of tRNA and for phosphorylation of GDP (Fig. 1A and S1).”.

(d) Results: (1) Line 76: Thus to inhibit the fluidity … . What about: Thus to decrease the leak through the membrane,

Corrected.

(2) Line 121: significant fluorescence: give numbers, like this it does not mean anything.

We modified the Fig. 2A, E, and G by adding a plot profile graph as inset. And, we changed the sentence as “After 6 hours, we observed the fluorescence of internally synthesized GFP by confocal microscopy (Fig. 2A).”.

(3) Figure 2F: what is the ratio between the two bands on the gel? The sample lacking bRFoF1-PLs (the right one) was used to confirm the position of GFP on the SDS-PAGE. We realized that this band does not mean a lot and brings confusion in this result, therefore we removed it and rearranged the figure.

(4) Line 188: and likely contributed: the authors do not seem sure of their claim.

We removed “likely”.

(4) Figure 3: it’s not a recursive system. It would be recursive if the light-driven regeneration provides the energy to synthesize the whole system (bacteriorhodopsin and ATP synthase, plus some other proteins like a reporter).

We changed the title of Fig. 3 as “**Self-constituting protein synthesis positive feedback-loop in artificial photosynthetic cells**”.

(e) Discussion: (1) Line 235: The chemical energy ... in ATP is a fundamental ...,

Corrected.

(2) Line 242: translated into a functional protein, GFP: not really, what's produced is only a small piece of GFP.

Corrected as "translated into a part of GFP".

(3) Line 249: It should be noted that all ...,

Corrected.

(4) Line 265: Our work demonstrated that ...: this work was already demonstrated by Lee et al in 2017.

We rewrote the sentence as "Our work demonstrated that a simple bio-system, which consists of two kinds of membrane proteins, is able to supply sufficient energy for operating gene expression inside a microcompartment.".

(f) other comments: a more convincing experiment would have been to synthesize the bacteriorhodopsin and ATP synthase using the ATP regeneration system added to the PURE (kinase and phosphate donor) and demonstrate that the synthesized light-driven system can take over the ATP regeneration to extend production of a reporter gene for days based on light only.

We thank the reviewer#2 for suggesting the interesting experiment idea. Indeed, it will be a more convincing result if we could show that both bR and FoF1 were synthesized in the cell-free system and the resulting light-driven system can take over the energy regeneration system. In order to explore whether we can practically approach this or not, we tried to synthesize whole subunit proteins of FoF1 in a standard PURE system including creatine phosphate and creatine kinase (CK). However, unfortunately, we could not observe the activity of the synthesized FoF1. On the other hand, when we synthesized Fo and F1 proteins separately in individual PURE system, then combined them after the reaction, we could observe the proton-pump activity of

the products. These results indicate that the productivity of PURE system does not reach to the detection limit of the FoF1 activity measurement. Another problem is that it is difficult to adjust the synthesis amount of the 8 subunit proteins to obtain the maximum FoF1 concentration. Additionally, we have found that an inhibitor, AP5A, for the CK does not completely block the CK activity in the system, which means that it is difficult to inactivate the primal energy regeneration system after the synthesis of light-driven system. Although we could not synthesize enough amount of FoF1, we knew that the PURE system can synthesize only Fo and make it functional by combining with purified F1 complex. Using this advantage, we synthesized 3 component proteins of Fo based on the photosynthesized ATP. To know if the synthesized Fo proteins integrated into the bRFoF1-PLs and enhanced its ATP synthesis activity, we synthesized the wildtype (a_{wt}) and mutant (a_{mut}) α -subunit (one of the 3 subunit proteins) of Fo and compared the ATP synthesis activity of both resulting bRFoF1-PLs, just like what we did in the bR photosynthesis experiment. As the result, we observed that the PLs-containing *de novo* F_o with a_{wt} showed higher ATP photosynthetic activity than the PLs-containing *de novo* F_o with a_{mut} in all three independent measurements. This data shows that photosynthesized Fo were integrated into the original bRFoF1-PLs and contributed into the photosynthetic activity.

A long-term protein synthesis is still difficult in the present condition because of a fundamental problem of PURE system that no one has succeeded so far. Therefore, we would like to try the long-term synthesis after waiting the activity improvement of the PURE system.

The supplementary information is composed of 4 tables, 17 figures, and the methods. Some comments:

(a) line 34: were (and not was).

Corrected.

(b) line 39: Aldrich.

Corrected.

(c) line 40: 45,000 g (same on line 45).

Corrected.

(d) line 41: more than 6 times, how many times really?

6 times. Corrected.

(e) line 49 and others: give complete coding sequence as text in the supplementary material for each gene cloned in this work, including promoter sequence.

We added Table S5 to provide DNA sequence information of the genes which cloned and used in this work.

(f) line 54: do you mean French press?

Corrected.

(g) for all the protein purified, specified the concentration of the stock solutions.

We added the sentences indicating the stock concentration of the purified proteins in the Material and Methods section.

(h) line 123: as deemed necessary, that's not a way to report scientific work, be more clear.

Delated.

(i) line 170: 16 mg/ml

Corrected.

(j) line 191: PURE buffer, which one?

We added a sentence as "PURE buffer, which is the same composition as GUV inside but without tRNAs".

(k) line 197: were (not was).

Corrected.

(l) line 220: 100,000

Corrected.

(m) spaces

?

(n) line 231: 8 mg/ml

Corrected.

(o) line 258: 157,800 g

Corrected.

(p) fig. S1: specify the kDa of bR.

We added a sentence as “The theoretical size of bR is 26.8 kDa.”.

(q) fig. S2: what are the units for the slopes in the table? Same for fig. S3.

We explained as “Relative Fluorescent Intensity/sec” in the figure and legend.

(r) fig. S7: specify the concentration of bR and FOF1 used.

We changed the sentences of Figure legend as “ATP synthesis rates (nM/min) were measured using PLs which consists of 146 μ M bR and 1 μ M F_oF₁. The PLs were illuminated for 10 min at the light intensities in 0, 0.8, 1.4, 4.5, 7.6, 14.3, 26.0, and 50.0 mW/cm².”.

(s) fig. S13: why are the signals for column 5 and 7 that high? They should be at background level.

This data shows the amount of synthesized GFP11, not fluorescent intensity of GFP. Because GFP11 has only 17 amino acids, it is difficult to confirm the product by gel. Therefore, we employed the filter binding assay and counted the radioactivities of the products by liquid scintillation counter.

(t) fig. S14: error bar, do you mean scale bar? What does ‘Non’ mean?

Corrected.

Reviewer #3 (Remarks to the Author):

The manuscript by Berhanu and colleagues describes construction of a synthetic minimal cell that can generate energy to sustain protein synthesis.

This is well designed and technically very sound paper.

This paper is touching on very similar subject to the recently published work: Lee, K. Y. et al. Photosynthetic artificial organelles sustain and control ATP-dependent reactions in a protocellular system. Nat. Biotechnol. (2018). doi:10.1038/nbt.4140

In that Nature Biotechnology paper, authors reconstituted ATP synthesis machinery embedded in a membrane of artificial synthetic cell organelle.

The present manuscript by Berhanu and colleagues is based on a very similar concept.

We thank the review#3 for giving positive comment. As the reviewer#3 suggested, we are using the same artificial photosynthesis system to produce ATP inside GUV. However, the main difference from the work by Lee *et al.* is that we used the produced energy for gene expression. Additionally, we showed that the photosynthesized proteins enhanced the activity of the original artificial organelle in the scheme of positive feedback loop. This is important in the aim of construction of self-reproducing artificial cells, which have to produce all molecules forming themselves. We would like to emphasize this point.

Minor points:

figure 1a is rather unclear. I get that the authors were aiming mostly for visual appeal here, but it's hard to understand what is going on.

We replaced the Figure 1A to the newly prepared figure that summarizes the design of our artificial photosynthetic cell. We also rewrote Fig. 1A legend.

The lipids used for giant unilamellar vesicle formation include both POPC and soyPC, which for the purpose of membrane formation are nearly the same thing. It would be interesting to note why authors decided to use those lipids, and also how was the composition of membrane decided. As in case of most membrane proteins, the

specific composition and size of membranes has usually big impact on measured activity.

We used POPC as the major lipid for giant unilamellar vesicle (GUV) formation, but SoyPC for the proteoliposome (artificial organelle) formation. The reason to use POPC is that we need a stable GUV where there are no proteins on the lipid membrane. On the other hand, as the reviewer#3 suggested, it has been known that only POPC is not sufficient for the function of membrane protein in many cases. For this reason, we used SoyPC for the formation of proteoliposome carrying bR and FoF1. Another merit of SoyPC is that these are stable in the reaction mixture of PURE system. These things were additionally described in the main part of manuscript.

In encapsulation experiments, there is usually large amount of liposomes that do not express genes from the encapsulated plasmids. Did authors ever investigate what percentage of “dark” liposomes observed in flow cytometry were “dark” because they did not encapsulate all necessary components, vs had all components but the circuit did not work for some reason? This would be important distinction if this technology is to be used for construction of more complex synthetic minimal cell.

As the Reviewer#3 suggested, we observed a certain percentage of the prepared GUVs were not showing fluorescence. Although we still do not have clear answer, several possibilities can be mentioned as following.

(A) The GUVs could not encapsulate enough amount of all components during the formation. Especially in the encapsulation of artificial photosynthetic organelle, according Lee et al. (Nat. Biotech. 2018), large size proteoliposome (200-1000nm) could not encapsulated efficiently or fused with the GUV membrane after the encapsulation, although small size ones (50-100nm) were well

encapsulated into the GUV. The size of our artificial organelle is mainly 100-200nm, so the efficiency might not be high as small size ones.

(B) Since we are using PURE system, which is a complex mixture, some translational factor might have been inactivated by aggregation with the oil that used for GUV preparation.

(C) The internally synthesized ATP might have leaked out to the exterior of GUV. The leak of molecules from inside to outside of GUV has been known so far, thus we are adding the same concentration of amino acids at the outside when we prepare GUVs.

We mentioned about the (A) in the main text of the manuscript.

What is the overall yield, or efficiency, of ATP synthesis? comparison to natural photosynthesis would be very useful for further development of this system.

We have tried to calculate and show the yield or efficiency of ATP photosynthesis by the bRF_0F_1 -PL in the results presented. But to estimate the yield of natural photosynthesis system and to compare that to our artificial system is not easy calculation, although it would be very useful for further development of our system as the reviewer#3 said. We would like to mention to this issue in the next research.

Reviewers' Comments:

Reviewer #1:

Remarks to the Author:

In the revised manuscript, the authors did a nice job responding to all concerns and questions raised by the three reviewers, which improved significantly the quality of the manuscript. The paper is also easier to read and understand. That said, I have two remaining concerns that I believe the authors need to address.

The first is with regards to GFP production in GUVs. I still don't think the authors correctly deliver an explanation on why only ~50% of GUVs make GFP. It doesn't make sense to me that the inefficiency of GFP production would stem from the fact that PURE contains multiple purified components. I would assume that the PURE system used in solution should be homogeneous no matter how many elements exist in the system. An example in this paper (<http://advances.sciencemag.org/content/advances/2/4/e1600056.full.pdf>) shows hundreds of water-in-oil droplets with μm size in diameter successfully produced three different fluorescent proteins in light-activated system. I presume the limiting factor of the low protein production would be the bR-PL's stability in GUV, i.e., the lifetime of bR-PL inside GUV. If bR-PL made of lipid bilayer will be merged into GUV with time, the bR-PL vesicle will fail to pump protons and the protein production will be not observed in the GUV. Might this be an explanation? I am just trying to understand.

In relation to my initial point 4, have the authors performed a statistical analysis? This is what I was intending when I used the word "significant." What is the result of such a test?

Reviewer #2:

Remarks to the Author:

NA

Reviewer #3:

Remarks to the Author:

The authors answered all my questions and they did all the necessary changes.

I think this is a really nice article and authors cleared my doubts about duplication of previous, similar work.

We thanks to the Reviewer#1 and Reviewer#3 for very kind comments to our revised manuscript.

The following are our responses to the Reviewer#1's concerns.

The first is with regards to GFP production in GUVs. I still don't think the authors correctly deliver an explanation on why only ~50% of GUVs make GFP. It doesn't make sense to me that the inefficiency of GFP production would stem from the fact that PURE contains multiple purified components. I would assume that the PURE system used in solution should be homogeneous no matter how many elements exist in the system. An example in this paper

(<http://advances.sciencemag.org/content/advances/2/4/e1600056.full.pdf>) shows hundreds of water-in-oil droplets with μm size in diameter successfully produced three different fluorescent proteins in light-activated system. I presume the limiting factor of the low protein production would be the bR-PL's stability in GUV, i.e., the lifetime of bR-PL inside GUV. If bR-PL made of lipid bilayer will be merged into GUV with time, the bR-PL vesicle will fail to pump protons and the protein production will be not observed in the GUV. Might this be an explanation? I am just trying to understand.

We agree that membrane fusion between internal PLs and GUV membranes has partially been happened during the photosynthesis reaction. Taking account this point, we added following sentence in the section of main text; “Additionally, we cannot deny the possibility that inactivity of the internal artificial organelle by the fusion of bRFoF1-PLs and GUV membranes is limiting the prolonged stability of artificial photosynthetic cell.”

In relation to my initial point 4, have the authors performed a statistical analysis? This is what I was intending when I used the word “significant.” What is the result of such a test?

We sorry that we forgot to show the p-values in Fig. 3. We did *t*-test using two-side for the data of Fig. 3E and 3H. The calculated *p*-value was 0.00015 and 0.004 for Fig. 3E and 3H, respectively. This was added in the legend of Fig. 3 as following, “***. $P < 0.01$, ***.

$P < 0.001$. P values were from two-side t -test.”.